# BSM: Small but Powerful Biological Sequence Model for Genes and Proteins

## Abstract

Modeling biological sequences such as DNA, RNA, and proteins is crucial for understanding complex processes like gene regulation and protein synthesis. However, most current models either focus on a single type or treat multiple types of data separately, limiting their ability to capture cross-modal relationships. We propose that by learning the relationships between these modalities, the model can enhance its understanding of each type. To address this, we introduce BSM, a small but powerful mixed-modal biological sequence foundation model, trained on three types of data: RefSeq, Gene Related Sequences, and interleaved biological sequences from the web. These datasets capture the genetic flow, gene-protein relationships, and the natural co-occurrence of diverse biological data, respectively. By training on mixed-modal data, BSM significantly enhances learning efficiency and cross-modal representation, outperforming models trained solely on unimodal data. With only 110M parameters, BSM achieves performance comparable to much larger models across both single-modal and mixed-modal tasks, and uniquely demonstrates in-context learning capability for mixed-modal tasks, which is absent in existing models. Further scaling to 270M parameters demonstrates even greater performance gains, highlighting the potential of BSM as a significant advancement in multimodal biological sequence modeling.

## 1 Introduction

Biological sequences—such as DNA, RNA, and proteins—are fundamental to an organism's functions, as they encode genetic information that determines structure, function, and regulatory mechanisms (Watson & Crick, 1953; Nirenberg & Matthaei, 1961). Understanding these sequences is vital for unraveling the mysteries of biological evolution, deciphering disease mechanisms, and elucidating molecular interactions.

By applying machine learning algorithms to large-scale biological sequence data, one can capture evolutionary effects and extract complex patterns in gene transcription and protein translation. This not only enhances our understanding of gene regulation and protein function but also enables the prediction and generation of complex biological functions, significantly advancing our comprehension of biology and life processes (Nguyen et al., 2024a).

Despite the rapid advancements in modeling biological sequences with machine learning, current efforts have primarily focused on creating unimodal models specialized for DNA, such as DNABert2 (Zhou et al., 2023), HyenaDNA (Nguyen et al., 2024b), Caduceus (Schiff et al., 2024), NT (Dalla-Torre et al., 2023); RNA, including RNA-FM (Chen et al., 2022); or proteins, like ESM2 (Lin et al., 2023), ProTrans (Ahmed et al., 2020), ProGen2 (Nijkamp et al., 2023). However, complex biological processes such as gene regulation, CRISPR immunity, and genetic transposition involve interactions across multiple modalities.

Recently, several methods have focused on developing models capable of handling both gene and protein data. For example, Evo (Nguyen et al., 2024a) is a 7B genomic foundation model pretrained on DNA sequences, which inherently contain the potential to express other modalities. It learns from large genomic regions to capture systems-wide interactions and enables the design of more sophisticated biological functions. LucaOne (He et al., 2024) is a 1.8B model that is trained on both gene and protein data separately, allowing the model to process and analyze both types of data concurrently. Although these models demonstrate excellent performance across various tasks,

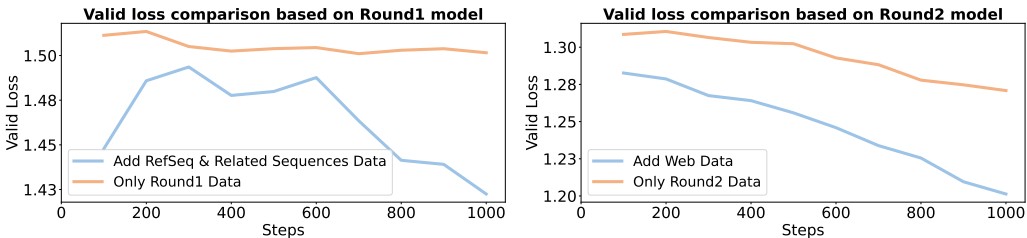

Figure 1: Impact of mixed-modal data on model learning efficiency. After the first round of training, the model shows slow learning efficiency when using only Round 1 data (single-modal data: DNA, RNA, and protein). However, when trained on the newly introduced mixed-modal data, it achieves significantly lower validation loss, indicating greatly improved learning efficiency. Similarly, the introduction of new web data after the second round further reduces validation loss. All validation losses across rounds are computed on the same Round 1 validation data for consistency.

they achieve their capabilities primarily by increasing computational resources—LucaOne, for instance, utilizes 800 billion tokens for pretraining—and by scaling up model size to the billions to accommodate multiple modalities. This raises the question of whether smaller models can achieve similar capabilities, making advanced biological sequence modeling more accessible and practical.

Recent advancements in small language models (SLMs) have shown significant progress (Minaee et al., 2024). These models have demonstrated emerging capabilities and achieved performance levels comparable to much larger models. The secret behind this achievement lies in strategic training choices, such as using "textbook-quality" data. Capable SLMs are faster to run and easier to serve; meanwhile, ongoing improvements, as seen with the Phi series (Abdin et al., 2024), indicate that SLMs are still under-trained and have great potential for further improvement.

In this paper, we explore the construction of high-quality biological sequence data. According to the central dogma of molecular biology (Crick, 1970), which highlights the sequential flow of genetic information, DNA, RNA, and proteins are intrinsically interconnected. We propose that by learning the relationships among these three types of sequences, the model can enhance its understanding of each modality. In light of this, we introduce three types of mixed-modal data for pretraining: RefSeq, Gene Related Sequences, and interleaved biological sequences from the web. These datasets capture genetic flow, gene and protein relationships, and the natural co-occurrence of diverse biological data types, respectively. Unlike previous work, which trains models using unimodal data or combines different types in a simplistic manner (where each training sample consists of only one type), we explicitly train the model on this mixed-modal data to learn the relationships among them.

We emphasize that incorporating mixed-modal data facilitates a more comprehensive understanding of biological sequences and enables more effective acquisition of cross-modal representations by better learning the relationships between these modalities. As illustrated in Figure 1, our experiments reveal that relying on unimodal data to learn these capabilities results in slow learning efficiency and requires substantial data and model sizes. In contrast, training on mixed-modal data significantly lowers validation loss, computed on the same Round 1 data across rounds, indicating greatly improved learning efficiency and better representation across all single modalities.

Based on these newly introduced types of high-quality mixed-modal data, we develop a small but powerful biological sequence foundation model, BSM, through a structured multi-round training approach. In this process, we conduct three rounds of training, progressively incorporating different types of mixed-modal data in the latter two rounds. By employing an annealing strategy, we optimize the data mix to ensure the best possible integration of these datasets. Our experiments demonstrate that BSM achieves performance comparable to that of billion-scale models on both single-modal and complex mixed-modal tasks. This proves the effectiveness of our method and its significant potential.

To conclude, our work makes the following contributions: 1) We propose that explicitly learning the relationships between genes and proteins can enhance the model's understanding of each modality. We introduce three types of mixed-modal data and strategically integrate them with unimodal data, ultimately resulting in our small but powerful BSM model. 2) We conduct extensive experiments

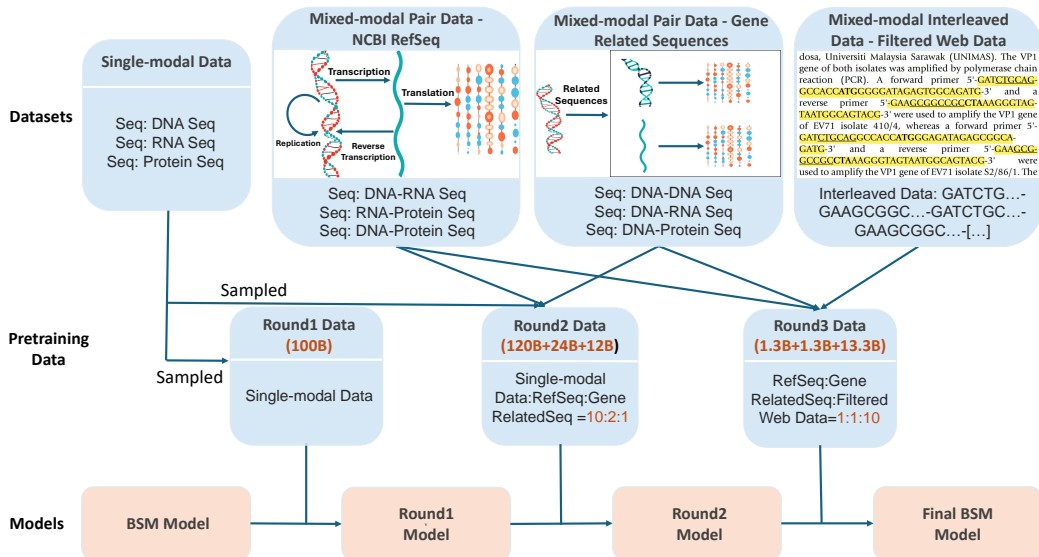

Figure 2: Overview of the pretraining data and training process of the BSM model. BSM utilizes three types of mixed-modal data: RefSeq, Gene Related Sequences, and interleaved biological sequences from the web for pretraining. It undergoes three rounds of training to enhance its ability to learn complex relationships among different types of biological data.

demonstrating that BSM achieves performance comparable to billion-parameter models across various tasks, particularly excelling in mixed-modal tasks, and exhibits unique and strong few-shot learning capabilities with different modality combinations. 3) We conduct scaling experiments to show that BSM scales effectively; when increased to 270M parameters, it achieves better results, highlighting the potential of our approach.

## 2 METHODS

### 2.1 ARCHITECTURE

BSM employs a single-nucleotide tokenizer with a vocabulary that includes nucleotides, amino acids, and special tokens. It uses an autoregressive architecture to model biological sequences such as genes and proteins. By learning next-token prediction, the model reasons over sequences causally and captures statistical patterns and dependencies in the training data, enabling effective representation and generation of biological sequences. Furthermore, the autoregressive architecture's sequential nature effectively handles long-range dependencies, which is crucial in biological sequences like DNA, RNA, and proteins, where long-context information can reveal critical functional relationships or structural interactions.

The BSM family includes models of two sizes, specifically BSM-110M and BSM-270M. BSM-110M is a decoder-only Transformer with 12 layers, each having 12 attention heads and a hidden dimension of 768, and BSM-270M features 20 layers, 16 attention heads, and a hidden dimension of 896. Both models utilize rotary position embedding (RoPE) (Su et al., 2024) with a base frequency hyperparameter of 100,000 and support a context length of 1024 tokens. To accelerate training, we employ flash-attention mechanisms (Dao, 2023).

### 2.2 PRETRAINING DATA

High-quality biological data play a key role in developing effective models for biological sequences. In addition to using unimodal protein and gene data, we incorporate three types of mixed-modal data for continued pretraining, each containing valuable information that helps the model learn diverse dependencies and interactions in biological sequences, as well as critical functional relationships.

These multi-modal datasets enhance the model's learning efficiency and its ability to understand cross-modal relationships, enabling it to handle various biological data. In the following section, we introduce the data used in pretraining. Details of the data construction are listed in the Appendix. B. Additionally, we perform data cleansing by removing instances of the downstream task test data from the pretraining data.

**Single-modal Data** In the initial pretraining phase, we utilize single-modal datasets that exclusively contain nucleic acid (DNA or RNA) sequences or protein sequences, enabling the model to concentrate on understanding the essential structures, patterns, and unique characteristics of each modality. These datasets are sampled from the pretraining dataset of LucaOne (He et al., 2024), and we only use the sequence information from RefSeq (O'Leary et al., 2016), UniProt (Consortium, 2015) and ColabFold (Mirdita et al., 2022) without incorporating additional biological annotations, resulting in a data volume of **220 billion** tokens. Single-modal data not only enhances the model's ability to understand individual modality sequences but also establishes a solid foundation for continued pretraining on more complex multimodal data.

**Mixed-modal Pair Data - NCBI RefSeq** To better learn the relationships among DNA, RNA, and proteins, we incorporate mixed-modal data from the NCBI RefSeq database (O'Leary et al., 2016). RefSeq provides a comprehensive and curated collection of annotated reference sequences that illustrate the flow of genetic information in the central dogma of molecular biology—DNA to RNA to protein. This resource is crucial for facilitating the understanding of gene-protein interactions and regulatory mechanisms, capturing essential details about transcription and translation processes.

We use data from 15 different species, including Bubalus Bubalis, Camelus Dromedarius, Human, and several others. Each species has its own unique DNA, RNA, and protein sequences, which are used to construct our gene-protein pairing dataset, resulting in a dataset of **9.2 billion** tokens. This extensive data ensures a rich representation of genetic information, allowing the model to leverage diverse biological contexts for effective learning while recognizing the inherent links between DNA sequences and their corresponding proteins.

**Mixed-modal Pair Data - Gene Related Sequences** To better understand the complex relationships between gene-gene and gene-protein, we incorporate data from the NCBI Gene database (Brown et al., 2015), which offers a detailed collection of gene-related sequences and annotations. The Gene Related Sequences data within this collection contains related sequences to the gene and provides links to the corresponding records in Entrez Nucleotide (Maglott et al., 2010),Entrez Protein (Ostell, 2012) or UniProtKB (Boutet et al., 2007).

We have constructed a dataset from the Gene Related Sequences data by sampling from multiple species, resulting in a diverse collection of **8.3 billion** tokens. This dataset enables the model to capture complex dependencies, recognize patterns of gene regulation and protein expression, and enhances its understanding of gene and protein functions.

**Mixed-modal Interleaved Data - Filtered Web Data** To simulate the natural co-occurrence of diverse biological data types and provide the model with a more realistic learning context, we integrate filtered web data from FineWeb-Edu (Penedo et al., 2024), which consists of high-quality web-crawled documents. We specifically filter this data to extract biological sequences within documents, using a special token, <sep>, to separate interleaved biological sequences, resulting in a final dataset of **33 million** tokens.

By incorporating this curated dataset into our pretraining process, BSM can leverage a rich and diverse collection of arbitrarily interleaved biological sequences. This dataset features a greater number of interleaved sequences, enhancing the model's ability to capture complex biological relationships and enabling it to understand and generate gene and protein sequences in any arbitrary context.

## 2.3 PRETRAINING PROCEDURE

**Three-round Training** We pretrain BSM models from scratch in an end-to-end manner, which includes a three-round training process, as shown in Figure 2. It begins by establishing a foundational understanding of individual types of biological sequences (DNA, RNA, or proteins) using 100B single-modal tokens. The model then advances to incorporate multi-modal data, enhancing its ability to understand relationships and transitions between different biological data types, which is crucial for tasks involving mixed modalities. Specifically, in the second round, it utilizes an additional 120B

single-modal data along with a certain amount of multi-modal pair data from RefSeq (24B) and Gene Related Sequences (12B). In the third round, it trains on a small amount of high-quality mixed-modal data, including pair data from RefSeq (1.3B) and Gene Related Sequences (1.3B) as well as web interleaved data (13.3B). By upsampling these high-quality but relatively small multi-modal datasets during continued pretraining, we significantly enhance the BSM model's performance across a range of biological tasks, making it a powerful tool for decoding the complexities of molecular biology.

We set the learning rate to decay from 2e-5 to 1e-5 in the first round. In the second round, it decays from 1e-5 to 1e-7, and in the third round, it decays from 1e-6 to 0. The tokenizer is trained with a peak learning rate of 2e-5.

**Simulated Annealing & Data Mixing** To obtain a high-quality biological model, it is essential to carefully determine the proportion of different data sources in pretraining. Similar to Blakeney et al. (2024) and Llama 3.1 (Dubey et al., 2024), we find that upsampling and annealing help efficiently select the optimal ratio for mixing new mixed-modal datasets. We evaluate these datasets by training the BSM-110M over 1000/500 steps with linearly annealed learning rates. Through these annealing experiments, we identify the best data mixing ratio based on the lowest validation loss in the second and third rounds. Ultimately, we employ a ratio of 10:2:1 for single-modal data, RefSeq, and Gene Related Sequences in the second round, and a ratio of 1:1:10 for RefSeq, Gene Related Sequences, and web interleaved data in the third round. After determining the best mix, we train a larger model (BSM-270M) on this selected data mix.

# 3 EXPERIMENTS

We evaluate BSM's capabilities in understanding and generating biological sequences across a variety of tasks, including both mixed-modal and single-modal tasks. BSM demonstrates outstanding performance on multi-modal tasks, even surpassing many billion-scale models. We also assess BSM's in-context learning (ICL) ability in a few-shot setting for mixed-modal tasks, demonstrating that BSM possesses this capability, which has not yet been observed in other biological sequence models. Additionally, we investigate the model's supervised fine-tuning (SFT) and zero-shot performance on protein and gene-related tasks. Scaling experiments confirm that further increasing the model size continues to enhance its performance. We conduct an ablation study on the performance of models from different rounds to verify the value of multi-modal data. Finally, we also evaluate the model's generative abilities using the perplexity metric. Details of evaluation datasets are listed in the Appendix C.

**Implementation Details** For tasks requiring SFT, we fine-tune the model using a learning rate of 1e-6 and a batch size of 16. For tasks that require two sequences as input, other baseline models lack the ability to simultaneously process both sequences, especially when it comes to handling gene and protein pairs. Instead, they use a dual-tower structure, employing two independent encoders to encode each sequence separately. In contrast, BSM directly connects the two sequences as input using a <sep> token, allowing the model to evaluate their relationship directly in a unified context. Implementation details are listed in the Appendix A.

## 3.1 MIXED-MODAL MODELING & FEW-SHOT EVALUATION

As shown in Figure 3, in mixed-modal tasks, such as RNA-protein interactions, BSM outperforms larger models like LucaOne. In the Central Dogma task, which focuses on DNA-protein associations, BSM achieves performance comparable to LucaOne. Additionally, in the few-shot learning setting without fine-tuning, BSM achieves performance close to SFT. Notably, BSM is the only existing biological sequence model capable of few-shot learning on mixed-modal data. These results highlight BSM's ability to efficiently process and analyze mixed-modal biological sequence data, positioning it as a leading model in the field despite its smaller size.

**ncRPI** The ncRNA-Protein Interactions (ncRPI) (Han & Zhang, 2023) task is a binary classification task aimed at predicting interactions between various non-coding RNAs (ncRNAs) and proteins, which is crucial for understanding cellular functions. Both BSM-110M and BSM-270M surpass the performance of billion-scale biological models, such as DNABert2 + ESM2-3B and LucaOne 1.8B. Notably, the results for BSM-270M are comparable to those of ncRPI-LGAT, a model specifically tailored for this task.

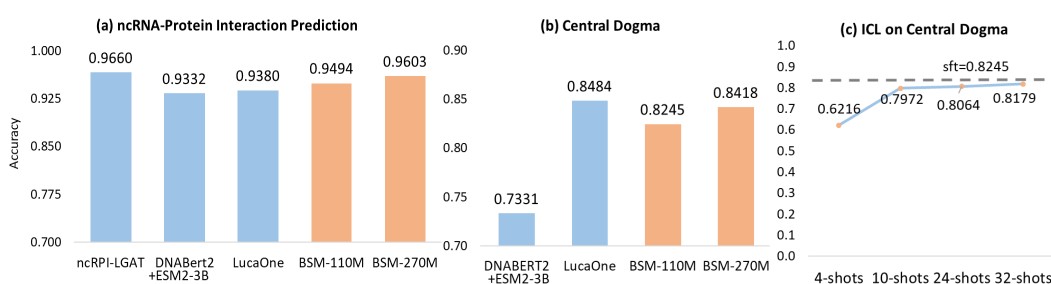

Figure 3: Results on mixed-modal tasks and few-shot evaluation. In the RNA-protein mixed-modal task (ncRPI), BSM outperforms larger models like LucaOne. In the DNA-protein mixed-modal task (Central Dogma), BSM achieves performance comparable to LucaOne. In few-shot learning settings without fine-tuning, BSM performs similarly to SFT, making it the only biological sequence model capable of few-shot learning on mixed-modal data.

**Central Dogma** The Central Dogma task is a binary classification task curated by LucaOne, aimed at recognizing the intrinsic association between DNA sequences and their corresponding proteins based on the central dogma. Specifically, the DNA-protein pairs are constructed from the RefSeq database (O'Leary et al., 2016). To ensure data integrity, we removed 57 instances of test data that were present in our pretraining dataset. We conducted two experimental settings for this task: one involved SFT with 3,200 training samples, while the other utilized few-shot learning without fine-tuning the model.

In the SFT experiment, BSM-270M performs comparably to LucaOne despite using a much smaller model size. Additionally, BSM outperform DNABERT2 + ESM2-3B in performance.

**Few-shot Learning** In the few-shot learning setting, we concatenate few-shot demonstrations with the test sample, each containing a DNA and protein sequence, as input for BSM. We then calculate the log probability for each token in the tested protein sequence. This allows us to obtain the overall generation probability for the tested protein sequence. We then set a threshold for this generation probability to classify whether the protein is linked to the tested DNA, and subsequently compute the prediction accuracy.

Despite not being fine-tuned, the BSM model demonstrates strong performance in identifying correct DNA-protein associations. The experiments show that increasing the number of demonstrations further enhances performance, with the few-shot learning results approaching those of SFT. This experiment highlights BSM's in-context learning capability, particularly in mixed-modal tasks, a capability that other existing models do not possess.

## 3.2 Protein Modeling Evaluation

We evaluate BSM's capabilities on four protein tasks, with results shown in Figure 4. Notably, we surpass all baseline models in both the PPI and ProtLoc tasks, achieving the best results. In the ProtStab task, we obtain results comparable to LucaOne. Additionally, in the zero-shot protein fitness prediction task, we achieve performance similar to Evo-7B and Progen2-large. These results highlight BSM's capability in modeling protein sequences through a deep understanding of protein functions and activities, despite its smaller size.

**PPI** The Protein-Protein Interaction (PPI) task is pivotal for mapping out how proteins interact within biological systems. We use the DeepPPI (Sun et al., 2017) database that contains human protein interactions for binary classification. The models are fine-tuned on this dataset, and their performances are assessed based on prediction accuracy. Both BSM-110M and BSM-270M surpass protein-specific models like DeepPPI and ESM2-3B, as well as multimodal biological sequence models like LucaOne.

**Prokaryotic Protein Subcellular Location (ProtLoc)** ProtLoc (Xu et al., 2009) predicts the subcellular localization of prokaryotic proteins, classifying them into six compartments like the cell membrane and cytoplasm. It uses a strategy similar to DeepLocPro (Moreno et al., 2024), helping to

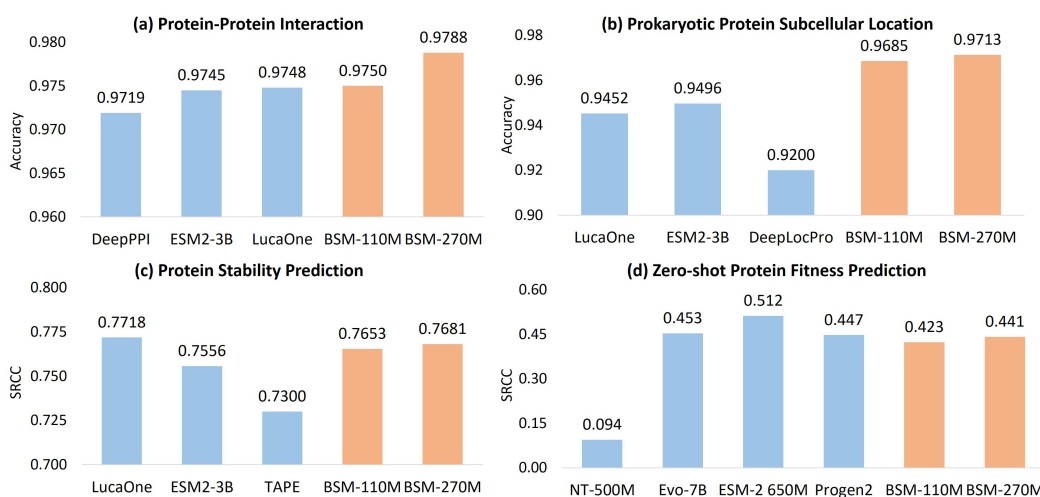

Figure 4: Results on four protein tasks. BSM outperforms all baseline models in PPI and ProtLoc, achieving the best results. In ProtStab, its performance matches LucaOne. Additionally, in the zero-shot protein fitness prediction task, BSM shows comparable results to Evo-7B and Progen2-large.

understand protein functions based on their locations. In this task, both BSM-110M and BSM-270M achieve the best results.

**Protein Stability (ProtStab)** evaluates protein stability by correlating features with stability measurements from the TAPE dataset (Rao et al., 2019). This task is important for understanding protein activity and can assist in drug design and biotechnology applications. In this task, BSM outperforms ESM-2-3B and TAPE, attaining results comparable to LucaOne.

**Zero-shot Protein Fitness Prediction** This task evaluates models' ability to predict the impact of mutations on protein function without task-specific fine-tuning. It uses Deep Mutational Scanning (DMS) datasets (Jacquier et al., 2013; Firnberg et al., 2014; Adkar et al., 2012; Tsuboyama et al., 2023; Kelsic et al., 2016), where a comprehensive set of mutations is introduced into protein-coding sequences to measure their effects on fitness. Fitness serves as a metric for how effectively a protein performs a specific function.

Following the implementation of Evo, the model predicts fitness scores based solely on its understanding of the protein sequence in a zero-shot setting. In the experiments, Evo-7B and NT-500M use gene sequences as input, while other protein models like ESM-2 650M and Progen2-large rely on protein sequences. In contrast, BSM utilizes both gene and protein sequences due to its mixed-modal modeling capability. Our experiments demonstrate that incorporating gene data enhances BSM-110M performance on this task, increasing the SRCC from 40.7% to 42.3%. This advancement not only establishes BSM's broad applicability in computational biology but also showcases its forward-looking nature in improving protein modeling capabilities through the integration of genetic information. Ultimately, BSM-270M achieves performance comparable to Evo-7B and Progen2-large, although it does not surpass ESM-2 650M.

### 3.3 GENE MODELING EVALUATION

We evaluate BSM's capabilities on several critical genomic challenges, with results shown in Figure 5. BSM outperformed Evo 7B in the zero-shot ncRNA fitness prediction task, leveraging its understanding of genomic sequences to predict the effects of mutations on ncRNA functionality without task-specific fine-tuning. It also performed well in the ncRNAFam multi-class classification task. These collective achievements underscore the model's comprehensive strength in genomic analysis and its potential to contribute significantly to molecular biology research.

**Zero-shot ncRNA Fitness Prediction** This task investigates the model's ability to predict the functional implications of mutations in non-coding RNAs (ncRNAs), including tRNAs, rRNAs,

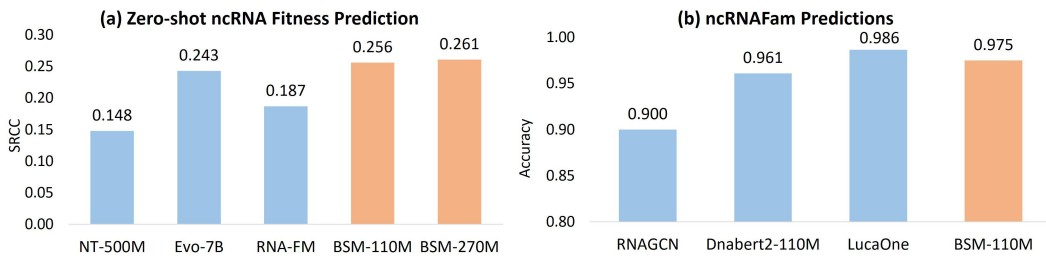

Figure 5: Results on two gene-related tasks. BSM outperformed Evo 7B in the zero-shot ncRNA fitness prediction task, accurately predicting the effects of mutations on ncRNA functionality without task-specific fine-tuning. It also performed well in the ncRNAFam multi-class classification task.

and ribozymes (Kobori et al., 2015; Andreasson et al., 2020; Domingo et al., 2018; Guy et al., 2014). Understanding these roles is crucial for cellular processes like protein synthesis and gene regulation. We use ncRNA Deep Mutational Scanning (DMS) data for evaluation, which includes various mutations and their effects on ncRNA functionality. Following Evo, we adopt a zero-shot approach to assess whether the pretrained BSM can generalize its understanding of genomic sequences to accurately predict the impact of mutations on ncRNA fitness without task-specific fine-tuning. Experimental results show that both BSM-110M and BSM-270M achieve the best performance, surpassing other methods, including Evo-7B.

**Non-coding RNA Family (ncRNAFam)** The ncRNAFam task is a sophisticated multi-class classification challenge, requiring models to accurately categorize non-coding RNA (ncRNA) sequences into 88 distinct families (Noviello et al., 2020; Rossi et al., 2019). These ncRNAs, while not coding for proteins, play indispensable roles in gene expression regulation and other cellular processes. Our fine-tuned BSM model achieves a remarkable accuracy of 97.5%, slightly below LucaOne but surpassing DNABert2 110M. This achievement underscores BSM's proficiency in discerning the subtleties of ncRNA sequences, showcasing its advanced capability to classify these crucial non-coding elements with high precision.

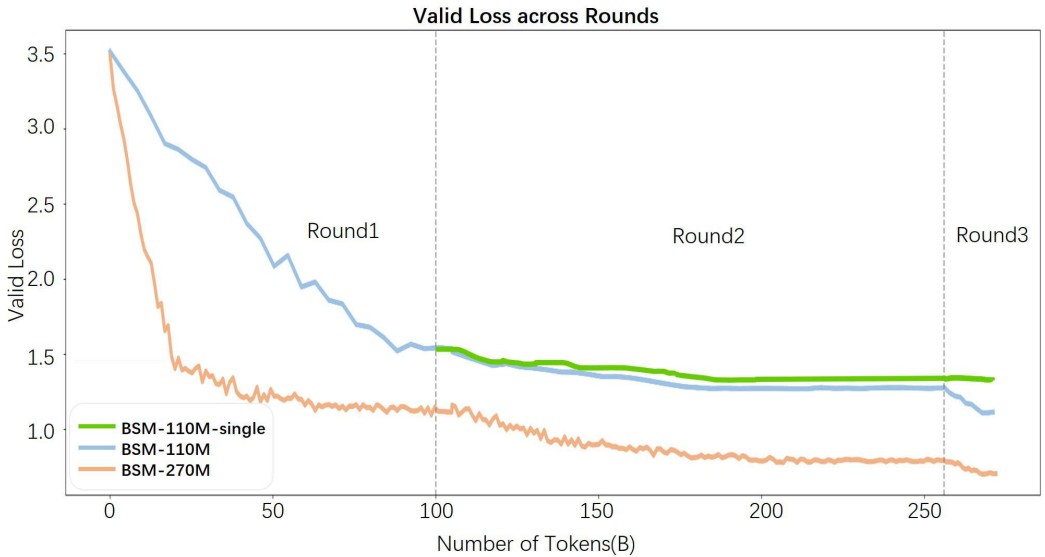

Figure 6: Effectiveness of scaling BSM and incorporating cross-modal data. Validation loss curves for the BSM-110M and BSM-270M show that scaling enhances model capabilities. Adding cross-modal data in Round 2 and Round 3 continuously reduces validation loss compared to the BSM-110M-single.

## 3.4 SCALING UP BSM

To unlock the full potential of BSM, we investigate its scaling properties by increasing its parameters from 110M to 270M. As shown in Figure 6, BSM-270M exhibits lower validation loss across all three rounds of training, demonstrating a significant improvement compared to BSM-110M. Experiments across diverse biological tasks also indicate that a larger model enhances performance. This scaling shows that increasing model size can further enhance BSM's capabilities, underscoring the effectiveness of our approach and the considerable value of incorporating mixed-modal data in advancing biological sequence modeling.

## 3.5 ABLATION STUDY ON MIXED-MODAL DATA

We compared models trained on single-modal versus with mixed-modal data under the same token budget on BSM-110M. The results in Table 1 show that BSM-110M (R2) outperforms BSM-110M-single (R2), and BSM-110M (R3) outperforms BSM-110M-single (R3). This demonstrates that incorporating mixed-modal data in both Round 2 and Round 3 leads to a significant improvement in model performance, both in single-modal and mixed-modal tasks. Additionally, without mixed-modal data, BSM-110M-single performs significantly worse than billion-scale models. However, when mixed-modal data is included, its performance matches or even exceeds that of these larger models. Figure 6 shows the validation loss of BSM-110M-single consistently higher than BSM-110M. This aligns with Figure 1 , confirming that introducing mixed-modal data significantly reduces validation loss and improves single-modal representations.

Table 1: Ablation study on mixed-modal data.

| Model | ncRPI | PPI | ProtLoc | Protein Fitness | ncRNA fitness |
|---|---|---|---|---|---|
| ESM2-3B | 0.9332 | 0.9745 | 0.9496 | / | / |
| LucaOne | 0.938 | 0.9748 | 0.9452 | / | / |
| Evo | / | / | / | 0.452 | 0.243 |
| BSM-110M-single (R2) | 0.9216 | 0.9648 | 0.9019 | 0.373 | 0.21 |
| BSM-110M (R2) | 0.9422 | 0.9722 | 0.9401 | 0.403 | 0.239 |
| BSM-110M-single (R3) | 0.922 | 0.9651 | 0.9038 | 0.379 | 0.211 |
| BSM-110M (R3) | 0.9494 | 0.975 | 0.9685 | 0.423 | 0.256 |

## 3.6 COMPARISON OF BSM WITH MODELS OF SIMILAR SIZE

We compared our models with ESM-150M, which share similar size with ours. Results in Table 2 show that the performance of ESM-150M is far lower than its larger models, and both BSM-110M and BSM-270M significantly outperform ESM-150M, highlighting the advantages of our approach and the importance of mixed-modal data. We clarify that we report ESM-650M for Zero-shot Protein Fitness Prediction because it is the best size for this task (Nguyen et al., 2024a).

Table 2: Comparison of BSM with ESM-150M of Similar Size

| Tasks | ESM-150M | ESM-650M | ESM-3B | BSM-110M | BSM-270M |
|---|---|---|---|---|---|
| PPI | 0.8139 | / | 0.9745 | 0.975 | 0.9788 |
| ProtLoc | 0.8644 | / | 0.9496 | 0.9685 | 0.9713 |
| Protein Stability | 0.7129 | / | 0.7556 | 0.7653 | 0.7681 |
| Protein Fitness | 0.408 | 0.512 | / | 0.423 | 0.441 |

## 3.7 PERPLEXITY EVALUATION

Perplexity (PPL) is one of the most common metrics for evaluating the generation capabilities of language models. It is defined as the exponentiated average negative log-likelihood of a sequence, reflecting how well a model can predict the next word based on the preceding context. A lower perplexity score indicates a better ability of the model to accurately predict the next word. We evaluate BSM's generation capability using perplexity on our validation protein data. As illustrated in Table 3, BSM outperforms the larger model ProGPT2 700M, although it doesn't surpass Progen2 2.7B. This demonstrates that using mixed-modal data for pretraining allows smaller models to effectively model and generate protein sequences.

Table 3: Comparison of BSM with various protein sequence models based on perplexity.

| Model | PPL. |
|---|---|
| Progen2 2.7B | 8.92 |
| Progpt2 700M | 9.75 |
| BSM-270M | 9.47 |

## 4 RELATED WORK

### 4.1 BIOLOGICAL SEQUENCE MODEL

Modeling biological sequences has traditionally involved unimodal approaches tailored to specific data types, such as DNA, RNA, or proteins. While significant progress has been made with models like DNABert2 (Zhou et al., 2023), RNA-FM (Chen et al., 2022), and ESM2 (Lin et al., 2023), these models often struggle with capturing complex mixed-modal interactions inherent in biological processes. Recent advancements, such as LucaOne (He et al., 2024) and Evo (Nguyen et al., 2024a), have begun to handle both gene and protein data, demonstrating the potential of large-scale models in modeling multi-modal biological data. However, insufficient attention has been given to exploring diverse data, especially high-quality mixed-modal data, which is crucial for models to acquire comprehensive capabilities. Our work proves that learning from mixed-modal data significantly enhances learning efficiency and improves both single and mixed-modal representations.

### 4.2 SMALL LANGUAGE MODEL

Small Language Models (SLMs) like Phi (Gunasekar et al., 2023) and Gemma 2 (Team et al., 2024) illustrate that with strategic training approaches, such as high-quality data utilization and knowledge distillation, SLMs can achieve impressive performance. Unlike current trends in biological sequence modeling that focus on scaling model size to the billion-parameter level, our research explores the potential of leveraging rich and high-quality mixed-modal bio-sequence data, which has rarely been studied or utilized in this field. We demonstrate that introducing mixed-modal data can enable smaller models to achieve performance close to or even surpass that of billion-scale models, highlighting the critical importance of data quality and diversity. Our work strongly demonstrates the tremendous potential of expanding both mixed-modal data and model size, paving the way for more powerful models in bioinformatics.

## 5 CONCLUSION

In this study, we have demonstrated that high-quality mixed-modal biological data is essential for enhancing both cross-modal and single-modal learning capabilities in our BSM models. The results indicate that protein-gene interleaving data has considerable potential to improve model performance, highlighting the importance of data quality in training effective biological models.

However, our research has certain limitations. We utilized only a partial dataset from RefSeq and Gene Related Sequence data, which lacks exploration of other valuable types of cross-modal data, such as gene-protein interactions data. Additionally, we mined only a relatively small dataset of interleaved biological sequences from the web, it still yielded continuous improvements in model performance. This suggests that there is substantial room for further investigation, and we believe that leveraging larger and more diverse datasets could enhance our model's capabilities even further. Our future work will focus on exploring additional types of cross-modal data to fully realize the potential of mixed-modal approaches in biological sequence modeling and contribute to advancements in the field.

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

## A   IMPLEMENTATION DETAILS

We use hyper-parameters listed in Table 4 for pretraining.

Table 4: Hyper-parameters for BSM-110M and BSM-270M models

| Hyper-parameter | BSM-110M | BSM-270M |
|---|---|---|
| Number of params (M) | 110 | 270 |
| Number of layers | 12 | 20 |
| Number of heads | 12 | 16 |
| Head dimensions | 768 | 896 |
| Context length | 1024 | 1024 |
| Batch size | 4096 | 1024 |
| Learning rate(R1/R2/R3) | 2e-5/1e-5/1e-6 | 2e-5/1e-5/1e-6 |
| Total steps | 32,500 | 124,000 |

## B   DETAILS OF PRETRAINING DATA CONSTRUCTION

**Mixed-modal Pair Data - NCBI RefSeq** We use 15 species data from the RefSeq dataset, which are listed in Table 5. We construct pair data using any combination of DNA, RNA, and protein, with a shuffled order.

Table 5: The 15 different species included in the RefSeq dataset utilized for pretraining phase.

| No. | Species | No. | Species | No. | Species |
|---|---|---|---|---|---|
| 1 | Bubalus Bubalis | 6 | Peromyscus Californicus | 11 | Mucor Saturninus |
| 2 | Camelus Dromedarius | 7 | Fusarium Annulatum | 12 | Penicillium Chermesinum |
| 3 | Human | 8 | Melampsora | 13 | Sporopachydermia Quercuum |
| 4 | Macaca Assamensis | 9 | Metschnikowia | 14 | Tranzscheliella Williamsii |
| 5 | Macaca Nigra | 10 | Mucor Saturninus | 15 | Xylariales |

**Mixed-modal Pair Data - Gene Related Sequences** We randomly extract 8.3 billion tokens from multiple species to construct our datasets, employing the same methodology for pairing DNA, RNA, and protein sequences in related sequences datasets, where the order is randomly selected.

**Mixed-modal Interleaved Data - Filtered Web Data** We filter documents from the Fineweb-Edu dataset that contain three or more biological sequences and use the extracted sequences to construct our dataset. This data ensures that the model is exposed to a richer variety of biological sequence data, simulating the natural co-occurrence of sequences in real bioinformatics environments.

## C   EVALUATION DATA DETAILS

**ncRPI** The ncRPI dataset is specifically designed for evaluating computational methods that predict interactions between non-coding RNA (ncRNA) and proteins (ncRPI). It consists of three sub-datasets: NPInter2.0, NPInter2.0_lncRNA, and RPI7317, covering thousands of experimentally validated ncRNA-protein interaction pairs identified from various model organisms. Specifically, the NPInter2.0 dataset contains 10,412 experimentally validated ncRNA-protein interaction pairs, involving 4,636 ncRNAs and 449 proteins; the NPInter2.0_lncRNA dataset includes 4,158 lncRNA-protein interaction pairs, involving 990 lncRNAs and 27 proteins; and the RPI7317 dataset contains 7,317 lncRNA-protein interaction pairs, involving 1,874 lncRNAs and 118 proteins. To generate negative samples (i.e., non-interaction pairs), researchers randomly paired ncRNAs and proteins from these datasets, creating an equal number of negative samples to the positive samples, ensuring that the total amount of positive and negative samples is equal during training and testing.

**Central Dogma** The central dogma dataset was meticulously designed and constructed to assess the model's ability to recognize the connection between DNA sequences and their corresponding proteins. The dataset selected 8,533 precise DNA-protein pairs from 13 species in the NCBI-RefSeq database. Each DNA sequence was extended to include an additional 100 nucleotides at both the 5' and 3' ends, with intron sequences preserved. To test the model's discrimination capabilities, researchers generated a number of negative samples double that of the positive samples by inserting, replacing, or deleting nucleotides in the DNA sequences or altering amino acids in the protein sequences. All samples were randomly allocated to the training, validation, and testing sets in a ratio of 4:3:25. The dataset is characterized by the inclusion of both positive samples and negative samples generated through various editing methods, which helps test whether the model can identify the intrinsic links between DNA sequences and corresponding proteins.

**PPI** The PPI dataset is a valuable resource for assessing protein-protein interactions (PPIs), sourced from the DeepPPI database, which includes a vast array of unique protein pairs. These datasets are crucial for unraveling the complex molecular communication mechanisms within cells. Specifically, the dataset comprises 59,766 training samples, 7,430 validation samples, and 7,425 test samples, collectively forming a binary classification task aimed at predicting whether a pair of protein sequences will interact. Each sample consists of a pair of protein sequences labeled as 1 (indicating interaction) or 0 (indicating no interaction). To ensure the accuracy and validity of the assessment, the dataset is meticulously divided to guarantee consistency in data distribution across the training, validation, and test sets. When constructing predictive models using this dataset, researchers must consider how to handle potential class imbalance issues and choose accuracy as the sole metric to measure model performance.

**ProtLoc** The ProtLoc dataset is a specialized dataset designed for predicting the subcellular localization of prokaryotic proteins. It comprises a curated selection of protein sequences from the UniProt and PSORTdb databases, with each sequence annotated for its specific subcellular location within the cell, such as the cytoplasm, cytoplasmic membrane, periplasmic space, outer membrane, cell wall and surface, and extracellular space, which are the six primary regions. This dataset is extensively used in the field of bioinformatics to train and evaluate machine learning models for the accurate prediction of protein subcellular localization. The dataset is divided into 9,915 training samples, 1,991 validation samples, and 1,131 test samples to support the training and assessment of models. When constructing predictive models using the ProtLoc dataset, accuracy is adopted as the primary evaluation metric to measure model performance.

**ProtStab** The ProtStab dataset is specifically designed for evaluating protein stability prediction models, sourced from the TAPE project, and includes stability measurement data for a variety of protein types, including natural proteins, mutants, and de novo designed proteins. The ProtStab dataset provides researchers with a benchmark for quantifying protein stability, aiding in the understanding of protein stability within biological systems and under different conditions. The dataset comprises 53,614 training samples, 2,512 validation samples, and 12,851 test samples, all of which support the training and evaluation of models. Each protein sample in the dataset is accompanied by a continuous numerical label indicating its stability measurement. When constructing and evaluating protein stability prediction models, Spearman's Rank Correlation Coefficient (SRCC) is used as the evaluation metric to measure the performance of the model.

**Zero-shot Protein Fitness Prediction** The Zero-shot Protein Fitness Prediction dataset is utilized for evaluating a model's ability to predict the impact of mutations on protein function without any task-specific fine-tuning or supervision. This dataset encompasses multiple Deep Mutational Scanning (DMS) studies, which include exhaustive mutation scans of protein-coding sequences, along with experimentally measured fitness scores that quantify the protein's ability to perform specific functions. The dataset comprises samples from both E. coli and human proteins, with the number of variants depending on the specific DMS study. Each sample consists of a protein sequence, the introduced nucleotide mutations, and the corresponding fitness score. The sequences are preprocessed to fit the input requirements of machine learning models. The performance of models on this dataset is assessed using the Spearman's Rank Correlation Coefficient (SRCC), which measures the strength and direction of association between the model's predictions and the experimental fitness measurements.

**Zero-shot ncRNA Fitness Prediction** The Zero-shot ncRNA Fitness Prediction dataset aims to evaluate a model's ability to predict the impact of mutations on the function of non-coding RNA (ncRNA) without any specific task fine-tuning or supervision. This dataset originates from multiple Deep

Mutational Scanning (DMS) studies that conduct exhaustive mutation scans on ncRNA sequences and measure the effects of these mutations on fitness through experimental means. Non-coding RNA plays a crucial role in many biological processes, including gene expression regulation, signal transduction, and cellular differentiation. The dataset includes samples from DMS studies, with each sample comprising the wild-type ncRNA sequence, the introduced mutations, and the corresponding fitness scores. The sequences are preprocessed to meet the input requirements of machine learning models. The performance of the model on this dataset is assessed using the Spearman's Rank Correlation Coefficient (SRCC), which measures the correlation between the model's predictions and the experimentally determined fitness scores.

**ncRNAFam** The ncRNAFam dataset is specifically designed for evaluating models that classify non-coding RNA (ncRNA) sequences into their respective families. Non-coding RNAs play a crucial role in many key biological processes, including the regulation of gene expression, signal transduction, and cellular differentiation. This dataset encompasses a variety of ncRNA sequence types, such as long non-coding RNAs and small nucleolar RNAs, each with its unique biological functions. It comprises 105,864 training samples, 17,324 validation samples, and 25,342 test samples, all of which support the training and evaluation of classification models. The dataset is roughly divided into 80% for training, 10% for validation, and 10% for testing. Each ncRNA sequence in the dataset is labeled according to its family classification, and the accuracy metric is used to assess the performance of the models.

Table 6: Evaluation data details.

| Task | Task Type | Input Type | Train/Valid/ Test Size | Seq Length (Max/Min/Mean) |
|------|-----------|------------|------------------------|---------------------------|
| ncRPI | Binary-Class(2) | RNA-Protein | 16,658/-/4,166 | 3,678/49/1,920 |
| Central Dogma | Binary-Class(2) | DNA-Protein | 3,200/2,400/20,000 | 617/11/244 |
| PPI | Binary-Class(2) | Protein-Protein | 59,766/7,430/7,425 | 33,423/24/593 |
| ProtLoc | Multi-Class(6) | Protein | 9,915/1,991/1,131 | 5,627/8/438 |
| ProtStab | Regression | Protein | 53,614/2,512/12,851 | 50/43/45 |
| ncRNAFam | Multi-Class(88) | RNA | 105,864/17,324/25,342 | 200/24/116 |

Additionally, Table 7 shows the species involved in different tasks.

We use Spearman Correlation Coefficient (SRCC) for the regression tasks, and Accuracy for the classification tasks. We followed Evo (Nguyen et al., 2024a) and LucaOne (He et al., 2024) in using the SRCC metric, which computes the Spearman correlation between the values (e.g., protein stability for ProtStab, fitness scores for proteins and ncRNAs) and the sequence likelihood (for autoregressive language models) or the sequence pseudolikelihood (for masked language models). Table 8 shows the details of label assignment for evaluation tasks.

Table 7: Species involved in different Tasks.

| Tasks | Species | Num. of species |
|---|---|---|
| **ncRPI** | Escherichia coli
Saccharomyces cerevisiae
Caenorhabditis elegans
Drosophila melanogaste
Mus musculus
Homo sapiens | 6 |
| **ProtLoc** | Archaea
Gram-positive bacteria
Gram-negative bacteria | 3 |
| **PPI** | Homo sapiens
Escherichia coli
Drosophila
Caenorhabditis elegans | 4 |
| **ProtStab** | / | / |
| **Zero-shot Protein Fitness Prediction** | Humans
Eukaryotes
Prokaryotes
Viruses | 4 |
| **Zero-shot ncRNA Fitness Prediction** | / | more than 10 |
| **ncRNAFam** | / | 41 |

Table 8: Details of label assignment for evaluation tasks

| Task | Task type | Label counts | Label description |
|---|---|---|---|
| Central dogma | Classification(2) | Train:1067(1)/2133(0); Test:6646(1)/13297(0) | Whether DNA seq translates to protein (1) or not (0) |
| ncRPI | Classification(2) | Train:8330(1)/8328(0); Test:2083(1)/2083(0) | Whether ncRNA-protein interact (1) or not (0) |
| PPI | Classification(2) | Train:33189(1)/26577(0); Test:4171(1)/3254(0) | Whether protein pair interacts (1) or not (0) |
| ProtLoc | Classification(6) | Train:2010(0)/75(1)/5913(2)/869(3)/592(4)/456(5); Test:325(0)/35(1)/250(2)/288(3)/160(4)/73(5) | Subcellular locations: cytoplasmic membrane(0), cell wall(1), cytoplasmic(2), extracellular(3), outer membrane(4), and periplasmic(5) |
| ncRNAFam | Classification(88) | Train: each class contains 1203 instances; Test: Max:4179(0),Min:1(62), Avg:287.9772 | The family classification of non-coding RNA (ncRNA) sequences |
| ProtStab | Regression | Train: Range: [-1.97, 3.40], Avg:0.1791, Counts: 21712(≤0)/31902(>0); Test: Range: [-1.16, 2.77], Avg:1.0020, Counts:59(≤0)/12792(>0) | The label represents a numerical value quantifying the intrinsic stability of each protein. |
| Protein Fitness | Regression | Train: Range: [-0.94, 2.46], Avg: 0.2649, Counts: 4070(≤0)/7215(>0); Test: Range: [-0.52, 2.31] Avg: 0.6872, Counts: 703(≤0)/3863(>0) | The fitness label reflects the effects of mutations on a protein sequence, measuring how well the protein performs a specific function. |
| ncRNA Fitness | Regression | Train: Range: [-1.52, 2.13], Avg: 0.4059, Counts: 1322(≤0)/3161(>0); Test: Range: [-1.16, 1.95], Avg: 0.6395, Counts: 318(≤0)/1313(>0) | The fitness label reflects the effects of mutations on non-coding RNAs. |

