# OpenReview forum: "BSM: Small but Powerful Biological Sequence Model for Genes and Proteins"
_ICLR.cc/2025/Conference — Submitted to ICLR 2025_

### Official Review · Reviewer_RZTE · 2024-10-31

**Soundness:** 2
**Presentation:** 3
**Contribution:** 2
**Rating:** 5
**Confidence:** 3

**Summary:**

This paper proposes to model the multiple biological sequences (DNA, RNA, and Protein) into a single model for cross-modal and uni-modal biological sequence tasks.
Authors propose to utilize small language model for the task, which outperforming larger models in various tasks.

**Strengths:**

- Modeling biological sequence and integrating heterogeneous types of sequences are important in building foundational model for biological sequences.
- Extensive experiments are conducted to demonstrate the superiority of BSM

**Weaknesses:**

- The main motivation of this paper is to use the small language model (SLM) for biological sequences. But why do we need SLM for biological sequence analysis? SLM is usually practical in real-life for implementing language model for on device or resource limited settings. However, in biological sequence analysis, scientists may already have intensive computational resources. It would be great if the authors can justify why we need SLM for biological sequence analysis.
- It may be convincing, if the SLM performs better than larger model due to the limited availability of data. However, as shown in various experiments, larger models perform better than smaller models. For example, Figure 3 (b) LucaOne performs better than BSM models, and BSM-270M model performs better than BSM-110M model.
- Experimental results
    - I think we need more ablation studies on multi modal data to further demonstrate the necessity of multi-modal pre-training approach.

**Questions:**

- How do you choose the data mixing ratio? Is it heuristically decided? How the model performance varies as mixing ratio changes?

---

> ### Author Response · Authors · 2024-11-22
> **Response to Reviewer RZTE (1/2)**
>
> Thank you very much for your valuable feedback. We address your questions point by point below.
>
> > Q1: The main motivation of this paper is to use the small language model (SLM) for biological sequences. But why do we need SLM for biological sequence analysis?
>
> We would like to emphasize that our main contribution lies in addressing an important but often overlooked issue in current research: **exploring rich and high-quality data, particularly cross-modal bio-sequence data, which has rarely been studied or utilized in this field**. Existing work has focused on scaling models to billions of parameters to achieve strong performance. In contrast, we demonstrate that **introducing cross-modal data can enable smaller models to achieve performance close to or even surpass that of billion-scale models**. This highlights the importance of data.
>
> Additionally, we show that scaling model size continues to improve performance with this data. Due to resource limitations, we were unable to further scale the model size, but our work strongly demonstrates the tremendous potential of expanding both cross-modal data and model size.
>
> > Q2: It may be convincing, if the SLM performs better than larger model due to the limited availability of data. However, as shown in various experiments, larger models perform better than smaller models. For example, Figure 3 (b) LucaOne performs better than BSM models, and BSM-270M model performs better than BSM-110M model.
>
> For the statement, "It may be convincing, if the SLM performs better than larger model due to the limited availability of data," we would like to clarify that this is not true. Our model used 270B tokens, while larger models like LucaOne-1.8B[1] used 800B tokens. Therefore, **we are approaching or surpassing these larger models not due to their use of less data, but because they did not use high-quality cross-modal data as we did**. This data can greatly enhance learning efficiency and model performance. In Q3, we added more experiments to support this conclusion.
>
> When we refer to large models, we mean models that are much larger than ours, such as Evo-7B[2], ESM-3B[3], and LucaOne 1.8B[1]. BSM-110M and BSM-270M are our own models, which are much smaller than the others. In comparison with these baselines, we conducted extensive experiments across 8 tasks, including gene, protein, and cross-modal tasks. Among these tasks, we achieved the best results in 4 tasks: ncRIP (note that ncRPI-LGAT is specifically tailored for this task and cannot be extended to others), PPI, ProtLoc, and Zero-shot ncRNA Fitness Prediction. For the remaining 4 tasks, although we did not achieve the best results, our performance was still very close to the best results.
>
> The fact that BSM-270M outperforms BSM-110M demonstrates that our method has scalability, and **continuing to scale model size with cross-modal data will lead to even stronger models**.
>
> Suggested by other reviewer, we compared our models with ESM-150M, which is similar in size to ours. The results show that **for models of the similar size, our performance is significantly higher**. Additionally, the baseline models we report for larger sizes perform much better than ESM-150M. By incorporating data, BSM-110M/270M can achieve performance close to or even exceeding that of much larger models, demonstrating the significant value of cross-modal data. (Added in `Section 3.6`)
>
> | Tasks                                | ESM-150M | ESM-650M | ESM-3B | BSM-110M | BSM-270M |
> |--------------------------------------|----------|----------|--------|----------|----------|
> | Protein-Protein Interaction          | 0.8139   | /        | 0.9745 | 0.975    | 0.9788   |
> | Prokaryotic Protein Subcellular Location | 0.8644   |    /      | 0.9496 | 0.9685   | 0.9713   |
> | Protein Stability Prediction         | 0.7129   |     /     | 0.7556 | 0.7653   | 0.7681   |
> | Zero-shot Protein Fitness Prediction | 0.408    | 0.512    | /      | 0.423    | 0.441    |
>
> > Q3: I think we need more ablation studies on multi modal data to further demonstrate the necessity of multi-modal pre-training approach.
>
> Thank you for your valueable suggestion. We ran additional experiments on the 110M model, training it on single-modality data with the same token budget as BSM_R2 and BSM_R3. The results show that BSM_R2 outperforms BSM_single_R2, and BSM_R3 outperforms BSM_single_R3. This confirms that performance gains come from incorporating cross-modal data. The results also show that **without cross-modal data, BSM_single_R3 is far lower than billion-scale models. However, with cross-modal data, it matches or exceeds their performance**. (Added in `Section 3.5`)

---

> ### Author Response · Authors · 2024-11-22
> **Response to Reviewer RZTE (2/2)**
>
> | Model                | ncRPI                       | PPI    | ncRNA fitness | protein fitness | ProtLoc |
> |----------------------|-----------------------------|--------|---------------|-----------------|---------|
> | ESM2-3B              | 0.9332 (DNABert2+ESM2-3B)    | 0.9745 | /             | /               | 0.9496  |
> | LucaOne              | 0.938                       | 0.9748 | /             | /               | 0.9452  |
> | Evo                  |              /               |    /    | 0.243         | 0.452           | /       |
> | BSM-single-R2 (256B) | 0.9216                      | 0.9648 | 0.21          | 0.373           | 0.9019  |
> | BSM-R2(256B)         | 0.9422                      | 0.9722 | 0.239         | 0.403           | 0.9401  |
> | BSM-single-R3 (272B) | 0.922                       | 0.9651 | 0.211         | 0.379           | 0.9038  |
> | BSM-R3(272B)         | 0.9494                      | 0.975  | 0.256         | 0.423           | 0.9685  |
>
> We also added the valid loss curve of BSM-110M-single to `Figure 6`, showing its valid loss consistently higher than BSM-110M. This aligns with `Figure 1`, confirming that introducing cross-modal data significantly reduces valid loss and improves single-modal representations.
>
> > Q4: How do you choose the data mixing ratio? Is it heuristically decided? How the model performance varies as mixing ratio changes?
>
> In Chapter 2.3, section "Simulated Annealing & Data Mixing," we provide a detailed explanation of how we use the Simulated Annealing technique [4] to select the optimal data mix ratio. We evaluate various data mixing ratios by training the BSM-110M model for 1000/500 steps, with the learning rate (lr) linearly annealed to 0 (Round 2: lr 1e-5→0, Round 3: lr 1e-6→0). The ratio with the lowest validation loss is selected as the final ratio.
>
> The table below shows the validation loss for different data mix ratios we recorded. Results indicate that including RefSeq in Round 2 reduces validation loss compared to excluding it, but further increasing its proportion (i.e., 10:4:0 vs. 10:2:0) does not improve performance, likely due to its limited size, where excessive sampling hurts results. Adding Gene Related Seq data (10:2:1) also improves performance. In Round 3, adding web data enhances performance, with further improvements observed as its weight is increased.
>
> |Round 2 Data Mix Ratio (Single-modal Data:RefSeq:Gene Related Seq)         | Valid Loss (1000 steps)|
> |------------------------|---------------|
> | 10:0:0     | 1.4991     |
> |10:2:1   | 1.4026       |
>
> | Round 3 Data Mix Ratio（RefSeq:Gene Related Seq:Filtered Web Data）         | Valid Loss (500 steps)   |
> |-------------------------------|---------------|
> | 2:2:1           | 1.2731        |
> | 1:1:1           | 1.2627        |
> | 1:1:2           | 1.2516        |
> | 1:1:10          | 1.2480        |
>
> ---
> [1] LucaOne: Generalized Biological Foundation Model with Unified Nucleic Acid and Protein Language
>
> [2] Sequence modeling and design from molecular to genome scale with Evo, Science 2024
>
> [3] Language models of protein sequences at the scale of evolution enable accurate structure prediction, Science 2023
>
> [4] The Llama3 Herd of Models
>
> &nbsp;
>
> We want to express our sincere gratitude for your review. If you have any further questions or concerns, please feel free to contact us at any time. **If our response has addressed your concerns, we sincerely hope you will consider raising your score.**
>
> Best regards,
> All Authors

---

> ### Author Response · Authors · 2024-11-25
> **Looking forward to your feedback**
>
> Dear Reviewer RZTE,
>
> We sincerely appreciate your constructive feedback on our manuscript. Guided by your insightful suggestions, we have strived to address each point thoughtfully, which we believe have significantly improved the quality of our work.
>
> Thank you once again for your time and valuable guidance. We're eager to hear your thoughts on these revisions.
>
> &nbsp;
>
> Best regards,
> All Authors

---

### Official Review · Reviewer_FwnU · 2024-11-03

**Soundness:** 2
**Presentation:** 1
**Contribution:** 2
**Rating:** 1
**Confidence:** 3

**Summary:**

This paper presents BSM, a compact mixed-modal biological sequence foundation model, trained using three types of data: RefSeq, Gene Related Sequences, and interleaved biological sequences from the web. The model's performance is assessed by comparing it to other existing larger models (e.g., LucaOne, ESM2, Evo, etc.) across various downstream tasks.

**Strengths:**

A compact mixed-modal biological sequence foundation model is introduced and assessed across various downstream tasks.

**Weaknesses:**

The clarity and precision of the paper could be greatly enhanced by making it more concise. For example:

•	The paper contains numerous repetitive phrases, sentences, and references that could be removed to improve readability.

•	Data and experimental details would be better summarized in tables for clarity and ease of comparison.

•	Section 4 is largely redundant.

•	Many figures are of poor quality, with barely readable legends.

Drawing conclusions from the experiments presented in the paper is challenging due to the lack of important details, including:

•	There are no details about the head used in the experiments in conjunction with the foundation models (FMs). These details are crucial and need to be carefully set for a head-to-head comparison with other FMs.

•	Additionally, for other FMs, there could be inverse scaling, where larger models do not consistently outperform smaller ones on all tasks (see McKenzie, et al. (2023). Inverse scaling: When bigger isn’t better. arXiv preprint arXiv:2306.09479). Therefore, it is difficult to conclude that the proposed BSM model performs the best, despite being the smallest, solely due to the introduction of additional data for training.

•	The differences in performance across models are mostly subtle in many experiments. Statistical analysis would be helpful here.

**Questions:**

See Weakness

---

> ### Author Response · Authors · 2024-11-22
> **Response to Reviewer FwnU (1/2)**
>
> Thank you very much for your valuable feedback. We address your questions point by point below.
> > Q1: Making paper more concise...Section 4 is largely redundant...Data and experimental details would be better summarized in tables for clarity and ease of comparison..
>
> Thank you for your suggestions. We have made some revisions based on your feedback. We have reduced the repetitive content in Section 4 and added some discussion, and will refine them further in the camera-ready version.
>
> We'd like to explain our choice to present the results using figures instead of tables. Our experiments cover 8 different tasks, including protein, gene, and various cross-modal tasks, each with its own baseline methods. It is challenging to merge these results into shared tables, and presenting them in too many separate small tables would not be effective. We will continue to optimize the presentation of results in future work.
>
> > Q2: There are no details about the head used in the experiments in conjunction with the foundation models (FMs). These details are crucial and need to be carefully set for a head-to-head comparison with other FMs.
>
> We apologize if we haven't fully understood your question. If by "head used in conjunction with the foundation models" you are referring to how two models are connected to handle tasks with two input sequences, these experimental results are reported by LucaOne [1], where they used a dual-tower structure. LucaOne conducted comprehensive experiments to compare different structures, and **we reported the best results of these structures**. In 3 out of the 8 tasks, a dual-tower structure is used for baseline models. We will include the final architecture used for each of these 3 tasks in the appendix in the future.
>
> Our BSM, on the other hand, has the capability to handle any number of sequences. We simply connect the two sequences as one input in BSM, so our experiments do not involve complex architectural choices.
>
> Finally, we would like to emphasize that our newly added ablation study, comparing performance **with and without cross-modal data using the same model structure and token budget**, shows significant benefits of adding cross-modal data. (Added in `Section 3.5`)
>
> | Model                | ncRPI                       | PPI    | ncRNA fitness | protein fitness | ProtLoc |
> |----------------------|-----------------------------|--------|---------------|-----------------|---------|
> | BSM-single | 0.922                       | 0.9651 | 0.211         | 0.379           | 0.9038  |
> | BSM        | 0.9494                      | 0.975  | 0.256         | 0.423           | 0.9685  |
>
> > Q3: Additionally, for other FMs, there could be inverse scaling, where larger models do not consistently outperform smaller ones on all tasks (see McKenzie, et al. (2023). Inverse scaling: When bigger isn’t better. arXiv preprint arXiv:2306.09479). Therefore, it is difficult to conclude that the proposed BSM model performs the best, despite being the smallest, solely due to the introduction of additional data for training.
>
> In all the baseline models we compared, some have only one size, such as LucaOne 1.8B and Evo 7B, while others have a family of models of different sizes. For those with models of different sizes, we want to emphasize that **we report the best-performing model for each task**.
>
> To further address your concern, we compared our models with ESM-150M, which is similar in size to ours. The results show that **for models of the similar size, our performance is significantly higher. Additionally, the baseline models we report for larger sizes perform much better than ESM-150M**. By incorporating data, BSM-110M/270M can achieve performance close to or even exceeding that of much larger models. (Added in `Section 3.6`)
>
> | Tasks                                | ESM-150M | ESM-650M | ESM-3B | BSM-110M | BSM-270M |
> |--------------------------------------|----------|----------|--------|----------|----------|
> | Protein-Protein Interaction          | 0.8139   | /        | 0.9745 | 0.975    | 0.9788   |
> | Prokaryotic Protein Subcellular Location | 0.8644   |    /      | 0.9496 | 0.9685   | 0.9713   |
> | Protein Stability Prediction         | 0.7129   |     /     | 0.7556 | 0.7653   | 0.7681   |
> | Zero-shot Protein Fitness Prediction | 0.408    | 0.512    | /      | 0.423    | 0.441    |

---

> ### Author Response · Authors · 2024-11-22
> **Response to Reviewer FwnU (2/2)**
>
> > Q4: The differences in performance across models are mostly subtle in many experiments. Statistical analysis would be helpful here.
>
> We have added variance to our experiments, while baseline results reported by [1,2] did not provide this information. Results prove the robustness and reliability of our experiments.
>
> (a) Protein-Protein Interaction
>
> | Models      | Accuracy    |
> |-------------|-------------|
> | DeepPPI     | 0.9719     |
> | ESM2-3B     | 0.9745      |
> | LucaOne     | 0.9748      |
> | BSM-110M    | 0.9750 ± 0.0017     |
> | BSM-270M    | 0.9788 ± 0.0026   |
>
> (b)Prokaryotic Protein Subcellular Location
>
> | Models      | Accuracy    |
> |-------------|-------------|
> | LucaOne     | 0.9452      |
> | ESM2-3B     | 0.9496      |
> | DeepLocPro  | 0.9200      |
> | BSM-110M    | 0.9685 ± 0.0028     |
> | BSM-270M    | 0.9713 ± 0.0041     |
>
> (c) Protein Stability Prediction
>
> | Models      | SRCC       |
> |-------------|-------------|
> | LucaOne     | 0.7718      |
> | ESM2-3B     | 0.7556      |
> | TAPE        | 0.7300      |
> | BSM-110M    | 0.7653 ± 0.0084     |
> | BSM-270M    | 0.7681 ± 0.0063     |
>
> (d)Zero-shot Protein Fitness Prediction
>
> | Models       | SRCC     |
> |--------------|----------|
> | NT-500M      | 0.094    |
> | Evo-7B       | 0.453    |
> | ESM-2 650M   | 0.512    |
> | Progen2      | 0.447    |
> | BSM-110M     | 0.423 ± 0.005   |
> | BSM-270M     | 0.441 ± 0.007  |
>
> We emphasize that our work is vital for advancing bio-sequence modeling. While current methods focus on scaling model size, we introduce a novel approach that greatly improves learning efficiency and model performance by incorporating cross-modal data. Our contribution is in presenting a new direction and proving its potential. Further exploration of cross-modal data and scaling models will lead to even stronger bio-sequence models.
>
> ---
> [1] LucaOne: Generalized Biological Foundation Model with Unified Nucleic Acid and Protein Language
>
> [2] Sequence modeling and design from molecular to genome scale with Evo, Science 2024
>
> [3] ProteinGym: Large-Scale Benchmarks for Protein Design and Fitness Prediction, NeurIPS 2023
>
> &nbsp;
>
> We want to express our sincere gratitude for your review. If you have any further questions or concerns, please feel free to contact us at any time. **If our response has addressed your concerns, we sincerely hope you will consider raising your score.**
>
> Best regards,
> All Authors

---

> > ### Comment · Reviewer_FwnU · 2024-12-02
> >
> > It is valuable to assess the importance of high-quality data on Foundation Models (FMs) and its impact on downstream tasks, and the effort to curate such data is commendable. However, to thoroughly analyze and conclude the significance of improvements resulting from data selection (the main contribution of this paper), a much more detailed experimental design and ablation study are required. This should include aspects such as the pre-training strategy (autoregressive vs. mask-based), the type of probing head and their hyperparameters (e.g., TextCNN vs. others), tokenization strategy, model architecture (e.g., transformer vs. SSM), and more, all of which significantly influence the final accuracy. These details are barely addressed in the current manuscript. Additionally, when the primary contribution is the selection of data for pre-training, it is crucial to analyze which types/categories of downstream tasks benefit the most from the proposed training data, as this benefit is unlikely to be universal. After carefully reading the authors' modifications and rebuttals, including their responses to other reviewers' comments, I am maintaining my score.

---

> > > ### Author Response · Authors · 2024-12-03
> > >
> > > > to thoroughly analyze and conclude the significance of improvements resulting from data selection (the main contribution of this paper), a much more detailed experimental design and ablation study are required. This should include aspects such as the pre-training strategy (autoregressive vs. mask-based), the type of probing head and their hyperparameters (e.g., TextCNN vs. others), tokenization strategy, model architecture (e.g., transformer vs. SSM), and more, all of which significantly influence the final accuracy.
> > >
> > > We have conducted extensive experiments and ablation studies to demonstrate the benefits of adding cross-modal data **under the same model structure and token budget**. The results are presented in `Table 1` and `Figure 6`:
> > >
> > > - Results in `Table 1` show that **without mixed-modal data, BSM-110M-single performs significantly worse than billion-scale models. However, when mixed-modal data is included, its performance matches or even exceeds that of these larger models**.
> > >
> > > - Results in `Figure 6`, using single-modal validation data, prove that cross-modal data in both R2 and R3 significantly reduces validation loss and enhances the representation of each individual modality.
> > >
> > > **As for the model structure and tokenizer choice you mentioned, these are not the focus of our work**. Our experiments have already shown that **BSM-single, without the introduction of cross-modal data, yields far worse results compared to other billion-scale models**. In contrast, BSM performs similarly to, or even surpasses, these models, **proving that the performance improvement comes from the data, not from other architectural choices**.
> > >
> > > Regarding the type of probing head, **we have already explained that our model does not require any special head design** you referred to. The gains from our method are not related to these head designs, which are needed only when baseline methods cannot handle multiple sequences simultaneously. LucaOne has performed extensive testing on the impact of different head choices, and we have reported the best results. If needed, we can include the details of the heads used in the baseline in a revised appendix.
> > >
> > > If the reviewer, despite claiming to have thoroughly reviewed our revisions and responses to other reviewers, still believes that our experiments do not adequately demonstrate that the gain comes from the newly added cross-modal data, **we would appreciate specific, constructive suggestions on how to design an ablation study that better isolates the effect of the new data**.
> > > **We do not agree with your suggestion to focus on "pre-training strategy (autoregressive vs. mask-based), the type of probing head and their hyperparameters (e.g., TextCNN vs. others), tokenization strategy, or model architecture (e.g., transformer vs. SSM)" as a reasonable ablation study design to prove the effectiveness of the data.**
> > >
> > > > Additionally, when the primary contribution is the selection of data for pre-training, it is crucial to analyze which types/categories of downstream tasks benefit the most from the proposed training data, as this benefit is unlikely to be universal.
> > >
> > > We have demonstrated improvements from adding cross-modal data across three types of tasks: single-modal (gene and protein) and mixed-modal tasks. In the newly added ablation study (`Figure 1`), the results of BSM-R2 vs. BSM-single-R2 and BSM-R3 vs. BSM-single-R3 across five tasks are clearly shown.
> > >
> > > In the additional experiments provided to reviewer kPoi, we split the validation data into gene and protein categories, and it was shown that adding cross-modal data reduces the loss on both single-modal data (i.e., gene and protein).
> > >
> > > | Round | Overall Valid Data | Protein Valid Data | Gene Valid Data |
> > > |-------|--------------------|--------------------|-----------------|
> > > | Round2 | 1.2769 | 1.3019 | 1.2727 |
> > > | Round3 | 1.2287 | 1.2588 | 1.2236 |
> > >
> > >
> > > We hope this information can, to some extent, address your concerns. If you have more specific suggestions or questions and can provide us with sufficient time to address them, we would be happy to offer further responses and resolve any issues.

---

> ### Author Response · Authors · 2024-11-25
> **Looking forward to your feedback**
>
> Dear Reviewer FwnU,
>
> We sincerely appreciate your constructive feedback on our manuscript. Guided by your insightful suggestions, we have strived to address each point thoughtfully, which we believe have significantly improved the quality of our work.
>
> Thank you once again for your time and valuable guidance. We're eager to hear your thoughts on these revisions.
>
> &nbsp;
>
> Best regards,
> All Authors

---

### Official Review · Reviewer_kPoi · 2024-11-08

**Soundness:** 2
**Presentation:** 3
**Contribution:** 2
**Rating:** 5
**Confidence:** 4

**Summary:**

The authors of this work present BSM, small biological sequence models that ostensibly achieve competitive performance with existing larger models through the virtue of combining:
1. Multiple types of data modalities (e.g DNA, RNA, and Protein sequences)
2. "Curating" the data used resulting in a smaller but higher quality dataset.

By being careful about the types and mixes of data used for pre-training, the authors are able to achieve performance competitive with much larger (e.g. 10x) larger models.

**Strengths:**

I like the general concept, there are myriad types of data that can be used for biological sequence modeling, and I'd like to believe that being smart about data/modality selection can lead to increased performance with fewer parameters/FLOPs by leveraging multiple types of information at once.

**Weaknesses:**

1. It is not clear what the exact effect of each phase (data mix) is.
    2. The BSM model is split up into three phases:
        3. Phase 1 trained on 100B single modal tokens (e.g each sequence is only nucleic acids or only protein sequences).
        4. Phase 2 is further trained on 146B tokens for 246B tokens seen in total
        5. Phase 3 is then further trained on $\sim$ 16.9B more tokens for 262.2B tokens seen in total.
    3. It is unclear to this reviewer what perfromance gains are due to the increased token budget vs from the different types of data included. For example, in Table 4, the authors attempt to show that increasing rounds (and therefore increasing token budgets) lead to a increase in downstream performance, but it's unclear if this gain is from the additional tokens or from the curration of the tokens. At the very least, a control where a BSM model is trained for the same number of tokens, but only with the tokens from phase 1 for example may help show that the performance increase is due to the incorporation of different types of data and not just due to training for longer.
    4. The authors kind of show something like this in Figure 1, but it's unclear to me exactly what the validation set is (is it data from that phase? Some fixed set of data sampling from phase 3? I don't think I ever found a detailed description in the paper) so I'm not sure if the authors are showing that incoporating the other phase data actually makes the loss decrease faster or if they're showing some sort of domain shift (e.g. if val is from phase 3 but they keep training on phase 2, we might expect the validation loss to not decrease as fast). Figure 6 suffers from a similar issue where I'm unsure of what the validation data and so I'm unsure how to interpret the changes in loss per phase.
    5. Taken together, all of these concerns lead me to doubt what the effect of the different data mixes actually is, which seems to be a key claim of the paper.
2. The Authors do not compare to the smaller models (e.g. the small ESM models) where applicable. ESM has available 650M, 150M, and even smaller models. On all the protein tasks, the authors only report ESM-3B, except for zs-Protein Fitness Prediction where they use ESM-650M. Unless I'm missing something, reporting their 110M and 270M compared to ESM-150M would help showcase how ther training setup and architecture compares to a model similar in size. Note: going down to ESM-650M only for that one task is odd, was this a placeholder or something and 3B was to be shown in the final submission or was there some other reason for only showing 650M?
3. There is some details missing on the setup of all the models evalauted. What sequence lengths are used, for models that are truncated (e.g. max of 1024 tokens), how are long sequences processed? What was the fine-tuning procedure for each of the models? How much does this type of processing affect different model performance? Architectural choices like these likely have some effect on the performance of each model observed, but it's currently unclear how much these types of decisions contributed to the observed performance and so it's difficult to disambiguate what the true effect of the data selection is.
4. More details are needed for each evaluation task. Appendix C and Table 4 are a good start, but there's details such as the number of instances of each class, the species the task was taken from, etc. Most of the tasks only report accuracy or SRCC, but it's unclear if this is an adequate summary statistic for each task given this missing information.

**Questions:**

Could the species you used for training also have an effect on downstream performance. Existing models usually train on human+mammal data, occasionally also incorporating data from invertebrates, bacteria, fungi, etc. I think the BSM models were trained on a mix of mammal and fungal DNA from RefSeq, was there any logic to the species selected? Is it possible that any differences in performance are (at least partly) due to the species selected? This may be a useful thing to point out for curation (i.e. we get better perf because we picked better species), but currently I can't actually tell what the affect of adding in each specie is nor how much each specie is represetned in the training data.  NOTE:  the assembly accession information for each specie accessed is missing from the paper, please make sure this is available somewhere since these assemblies tend to be updated periodically by NCBI.

How exactly were data leaks prevented? I understand that data from the downstream tasks was removed from pre-training but how exactly was this done and how are you sure that no leakage occurred by e.g. a very similar sequence, or having a gene whose transcribed protein is in the test set?

---

> ### Author Response · Authors · 2024-11-22
> **Response to Reviewer kPoi (1/2)**
>
> Thank you very much for your valuable feedback; your input is very helpful in improving our work. We address your questions point by point below.
>
> > Q1: It is unclear to this reviewer what performance gains are due to the increased token budget vs from the different types of data included.
>
> > Q2:The authors kind of show something like this in Figure 1, but it's unclear to me exactly what the validation set is.
>
> To address them, we first clarify the valid loss is computed on validation data from Round 1, which includes only single-modality data (DNA, RNA, and protein). **The valid losses across all rounds are computed on the same data for consistency.** This was explained in `Figure 1`’s caption and `Line 95` of the original paper, and we have now revised the `PDF` to make this clearer, with changes highlighted in blue.
>
> `Figure 1` shows that introducing new cross-modal data in Round 2 and Round 3 significantly reduces valid loss compared to not adding such data, indicating improved representation across all single modalities.
>
> To address Q1, we ran additional experiments on the 110M model, training it on single-modality data with the same token budget as BSM_R2 and BSM_R3. The results show that BSM_R2 outperforms BSM_single_R2, and BSM_R3 outperforms BSM_single_R3. This confirms that performance gains come from incorporating cross-modal data rather than longer training. The results also show that without cross-modal data, **BSM_single_R3 is far lower than billion-scale models. However, with cross-modal data, it matches or exceeds their performance.** (Added in `Section 3.5`)
>
> | Model                | ncRPI                       | PPI    | ncRNA fitness | protein fitness | ProtLoc |
> |----------------------|-----------------------------|--------|---------------|-----------------|---------|
> | ESM2-3B              | 0.9332 (DNABert2+ESM2-3B)    | 0.9745 | /             | /               | 0.9496  |
> | LucaOne              | 0.938                       | 0.9748 | /             | /               | 0.9452  |
> | Evo                  |              /               |    /    | 0.243         | 0.452           | /       |
> | BSM-single-R2 (256B) | 0.9216                      | 0.9648 | 0.21          | 0.373           | 0.9019  |
> | BSM-R2(256B)         | 0.9422                      | 0.9722 | 0.239         | 0.403           | 0.9401  |
> | BSM-single-R3 (272B) | 0.922                       | 0.9651 | 0.211         | 0.379           | 0.9038  |
> | BSM-R3(272B)         | 0.9494                      | 0.975  | 0.256         | 0.423           | 0.9685  |
>
> We also added the valid loss curve of BSM-110M-single to `Figure 6`, showing its valid loss consistently higher than BSM-110M. This aligns with Figure 1, confirming that introducing cross-modal data significantly reduces valid loss and improves single-modal representations.
>
> > Q3: The Authors do not compare to the smaller models (e.g. the small ESM models) where applicable. Reporting their 110M and 270M compared to ESM-150M would help showcase how ther training setup and architecture compares to a model similar in size.
>
> > Q4: On all the protein tasks, the authors only report ESM-3B, except for zs-Protein Fitness Prediction where they use ESM-650M. Note: going down to ESM-650M only for that one task is odd.
>
> For Q4, we report ESM-650M for zs-Protein Fitness Prediction because it is the best size for this task. We follow Evo [1], which provides a clear explanation: "We also included ESM 2 650M and ProGen2 large given that these models have sometimes shown better performance at function prediction than larger variants of these models (Notin et al., 2023) [2]."
>
> For Q3, thank you for the suggestion. We compared our models with ESM-150M, which share similar size with ours. Results show that **ESM-150M is far lower than its larger model, and both BSM-110M and BSM-270M significantly outperform ESM-150M**, highlighting the advantages of our approach and the importance of cross-modal data. (Added in `Section 3.6`)
>
> | Tasks                                | ESM-150M | ESM-650M | ESM-3B | BSM-110M | BSM-270M |
> |--------------------------------------|----------|----------|--------|----------|----------|
> | Protein-Protein Interaction          | 0.8139   | /        | 0.9745 | 0.975    | 0.9788   |
> | Prokaryotic Protein Subcellular Location | 0.8644   |    /      | 0.9496 | 0.9685   | 0.9713   |
> | Protein Stability Prediction         | 0.7129   |     /     | 0.7556 | 0.7653   | 0.7681   |
> | Zero-shot Protein Fitness Prediction | 0.408    | 0.512    | /      | 0.423    | 0.441    |

---

> ### Author Response · Authors · 2024-11-22
> **Response to Reviewer kPoi (2/2)**
>
> > Q5: What sequence lengths are used, for models that are truncated, how are long sequences processed?
>
> The BSM model supports a maximum context length of 1024 tokens (`Line154` & `Table4`). Sequences longer than this are truncated.
>
> > Q6: What was the fine-tuning procedure for each of the models? How much does this type of processing affect different model performance? Architectural choices like these likely have some effect on the performance of each model observed, but it's currently unclear how much these types of decisions contributed to the observed performance and so it's difficult to disambiguate what the true effect of the data selection is.
>
> We apologize for not being entirely clear on what is meant by the fine-tuning procedure here. If you're referring to the baseline models using a dual-tower structure for tasks that require two sequences as input, these experimental results are reported by LucaOne[3]. LucaOne conducted comprehensive experiments to compare different architectures, and **we reported the best results of these architectures**. In 3 out of the 8 tasks, a dual-tower structure is used for baseline models. We will include the final architecture used for each of these 3 tasks in the appendix in the future.
>
> Our BSM, on the other hand, has the capability to handle any number of sequences. We simply connect the two sequences as one input in BSM, so our experiments do not involve complex architectural choices.
>
> **Finally, we would like to emphasize that the ablation experiment, conducted to answer Q1 above with the same model structure and token budget, shows significant benefits of adding cross-modal data.**
>
> > Q7: More details are needed for each evaluation task. Appendix C and Table 4 are a good start, but there's details such as the number of instances of each class, the species the task was taken from, etc.
>
> > Q8: Most of the tasks only report accuracy or SRCC, but it's unclear if this is an adequate summary statistic for each task given this missing information.
>
> For Q7, we have added the species information for the tasks in `Appendix C`. But we can't find the number of instances for each class in any related work.
>
> For Q8, to maintain comparability with previous work, we used the same metrics as in [1][3], which are widely used.
>
> > Q9: Could the species you used for training also have an effect on downstream performance. Was there any logic to the species selected? Is it possible that any differences in performance are (at least partly) due to the species selected?
>
> We used a mix of mammals (including humans) and fungi for our RefSeq species. The specific species are listed in `Table 5` (old pdf Table 4). The species were randomly selected during data collection and were not specifically designed for downstream tasks. Our experiments include 8 diverse tasks from a wide range of species, including mammals, invertebrates, bacteria, viruses, prokaryotes, and more. So we believe our results are reliable.
>
> > Q10: How exactly were data leaks prevented? I understand that data from the downstream tasks was removed from pre-training but how exactly was this done and how are you sure that no leakage occurred by e.g. a very similar sequence, or having a gene whose transcribed protein is in the test set?
>
> We removed sequences from the pretraining data that appear in the test set, without considering similar sequences like those involved in transcription. Please note that unlike LucaOne, **our training only uses sequence data**, without attribute data. For the 8 downstream tasks, **the label information for 7 tasks (except Central Dogma) is not included in the training data**. Five of these tasks are classification or regression, and the other two are ncRNA-Protein and Protein-Protein Interactions, which are also not included in the pretraining data. We don't think only using sequence data in pretraining would cause leakage for these tasks. However, we did remove these test sequences, as we did for the Central Dogma task. Therefore, we believe our results are fully reliable.
>
> ---
> [1] Sequence modeling and design from molecular to genome scale with Evo, Science 2024
>
> [2] ProteinGym: Large-Scale Benchmarks for Protein Design and Fitness Prediction, NeurIPS 2023
>
> [3] LucaOne: Generalized Biological Foundation Model with Unified Nucleic Acid and Protein Language
>
> &nbsp;
>
> We want to express our sincere gratitude for your review. If you have any further questions or concerns, please feel free to contact us at any time. **If our response has addressed your concerns, we sincerely hope you will consider raising your score.**
>
>
> Best regards,
> All Authors

---

> > ### Comment · Reviewer_kPoi · 2024-11-24
> > **Unclear task statistics and concerns over data leakage**
> >
> > Thank you for your response!
> >
> > You addressed enough of my concerns that I'm considering raising my score, but there are still a few concerns I have prevent me from doing so.
> >
> > ## Major Concern
> > * For Q7, we have added the species information for the tasks in Appendix C. But we can't find the number of instances for each class in any related work.
> >  > I may be wrong, but don't you have some list/file that contains the labels for each datapoint for each task? I was just asking for the value counts for each of classification tasks, and maybe some summary statistics for the regression tasks since I'm not familiar with all of the tasks you use. I understand you're using the same metrics as previous work, and I'm giving the tasks from previous work the benefit of the doubt that they're evaluated properly, but I would have imagined this is something pretty easy to calculate since you ran the evaluation of all your models on these tasks. I hope this was just a miscommunication during the review process, but if the authors mean to tell me that they are unable to calculate statistics like this, that would indicate to me some serious methodological problems with your evaluation setup and I am unable to increase my score at this time as a result.
> >
> > If the authors can address this adequately, their rest of their response was enough to convince me to up the score a bit, but I still have some concerns/points of confusion listed below.
> >
> > ## Other concerns
> >
> > You've addressed my concerns about the data mixing being useful, but I still have some concerns about it, namely:
> > 1. Did the authors sample new tokens to continue training or did they use the  data as phase 1 to continue training? I'm sorry if I missed this somewhere in the new work, but my concern is that if their experiment used the data as phase 1 (that is repeating sequences) then I would expect a marginal gain at best in model performance. If the authors sampled a new set of data from phase 1 (eliminating or minimizing the overlap with the original phase 1 data), then this helps alleviate the concern, although I'm still a bit unclear of the exact phase make-up for each phase (see concern 3).
> > 2. Why is the phase 3 data so effective?  Phase 3 (the web data) seems to have  a **dramatic** impact in a relatively few number of tokens. My concern is that the **web data is naturally biased towards sequences of interest that might show up in a dataset** (e.g. protein/genes being studied for some reason) and the datasets you use to evaluate are likely contain very similar proteins to those (e.g. sequences that are not identical but with a small edit distance, or the DNA sequence of a protein that's in one of your test sets)). I'm concerned that what you're showing isn't that "varied forms of data help", but rather that "pre-training on a set of sequences similar to those we find in the test set improves performances". While you've convinced me you're not leaking labels, I'm still concerned that your model is seeing very similar sequences in the pre-training data particularly in phase 3. The val ppl also decreasing rapidly may help point that phase 3 data actually does help the model learn something useful, but I'm not sure you can quite say this without a more rigorous exploration of what is in the validation data on how it affects different parts of it (e.g. does it decrease the ppl in only the protein portion of the val or uniformly across them)?.  Perhaps I'm misunderstanding something about this and would appreciate  some clarification on whether this is actually happening in practice or whether my concern is unfounded.  I think this is worth discussion in much more detail in the main paper (what aspect of the web data is so effective and why are we sure our gains aren't from accidents leakage). Additionally, assuming that the web data is fine, why even bother with the first 2 phases then?
> > 3. I'm still unclear on the mix of phase 1 data used for training and eval. What's the mix of DNA:RNA:Protein, how was this data sampled? (uniformly at random? Sampling specific chromosomes/genes/ etc for some set of species?)  I assume the authors also did some filtering/selection to make sure the val data didn't show up in the training data for each different phase, was the some process as the test data filtering used? This is the short of granularity I was looking for when I asked for a description  of the val data, and given how data curation seems to be such a important aspect of this paper I also expected this for basically every phase too. To get your same performance gains is sampling uniformly at random from the same databases sufficient, or was there more that went into the sampling? Are they gains form each phase really from the multi-modal aspect of them or did they just happen to match the distribution of DNA:RNA:Protein better than your phase 1 data?

---

> ### Author Response · Authors · 2024-11-25
> **Response to Address Reviewer kPoi's Additional Questions (1/4)**
>
> Thank you again for your highly valuable feedback. As with the last feedback, your input is very helpful in improving our work.
>
> > I may be wrong, but don't you have some list/file that contains the labels for each datapoint for each task? I was just asking for the value counts for each of classification tasks, and maybe some summary statistics for the regression tasks since I'm not familiar with all of the tasks you use. I understand you're using the same metrics as previous work, and I'm giving the tasks from previous work the benefit of the doubt that they're evaluated properly, but I would have imagined this is something pretty easy to calculate since you ran the evaluation of all your models on these tasks. I hope this was just a miscommunication during the review process, but if the authors mean to tell me that they are unable to calculate statistics like this, that would indicate to me some serious methodological problems with your evaluation setup and I am unable to increase my score at this time as a result.
>
> We sincerely apologize for not providing you with this information earlier. As you mentioned, we can calculate this information. We have added the calculated information to the Table 1 below and included it in `Appendix C`. If you would like to know any information beyond this table, please let us know.
>
> We use Spearman Correlation Coefficient (SRCC) for the regression tasks, and Accuracy for the classification tasks. We followed LucaOne/Evo [1,2] in using the SRCC metric, which computes the Spearman correlation between the values (e.g., protein stability for ProtStab, fitness scores for proteins and ncRNAs) and the sequence likelihood (for autoregressive language models) or the sequence pseudolikelihood (for masked language models).
>
> Table 1:
>
> | Task     | Task type | Value counts | Label description |
> |----------|-----------|--------------|-------------------|
> | Central dogma | Classification(2) | Train:1067(1)/2133(0); Test:6646(1)/13297(0) | Whether DNA seq translates to protein (1) or not (0) |
> | ncRPI    | Classification(2) | Train:8330(1)/8328(0); Test:2083(1)/2083(0) | Whether ncRNA-protein interact (1) or not (0) |
> | PPI      | Classification(2) | Train:33189(1)/26577(0); Test:4171(1)/3254(0) | Whether protein pair interacts (1) or not (0) |
> | ProtLoc  | Classification(6) | Train:2010(0)/75(1)/5913(2)/869(3)/592(4)/456(5); Test:325(0)/35(1)/250(2)/288(3)/160(4)/73(5) | Subcellular locations: cytoplasmic membrane(0), cell wall(1), cytoplasmic(2), extracellular(3), outer membrane(4), and periplasmic(5) |
> | ncRNAFam | Classification(88) | Train: each class contains 1203 instances; Test: Max:4179(0), Min:1(62), Avg:287.9772 | The family classification of non-coding RNA (ncRNA) sequences. |
> | ProtStab | Regression | Train: Range: [-1.97, 3.40], Avg:0.1791, Counts: 21712(≤0)/31902(>0); Test: Range: [-1.16, 2.77], Avg:1.0020, Counts:59(≤0)/12792(>0) | The label represents a numerical value quantifying the intrinsic stability of each protein. |
> | Protein Fitness | Regression | Train: Range: [-0.94, 2.46], Avg: 0.2649; Counts: 4070(≤0)/7215(>0); Test: Range: [-0.52, 2.31], Avg: 0.6872; Counts: 703(≤0)/3863(>0) | The fitness label reflects the effects of mutations on a protein sequence, measuring how well the protein performs a specific function. |
> | ncRNA Fitness | Regression | Train: Range: [-1.52, 2.13], Avg: 0.4059; Counts: 1322(≤0)/3161(>0); Test: Range: [-1.16, 1.95], Avg: 0.6395; Counts: 318(≤0)/1313(>0) | The fitness label reflects the effects of mutations on noncoding RNAs. |
>
> Regarding your other three new concerns, we will address them one by one as follows.
>
> > 1. Did the authors sample new tokens to continue training or did they use the data as phase 1 to continue training? I'm sorry if I missed this somewhere in the new work, but my concern is that if their experiment used the data as phase 1 (that is repeating sequences) then I would expect a marginal gain at best in model performance. If the authors sampled a new set of data from phase 1 (eliminating or minimizing the overlap with the original phase 1 data), then this helps alleviate the concern, although I'm still a bit unclear of the exact phase make-up for each phase (see concern 3).
>
> We previously sampled a total of 220B single modal data from LucaOne [1] (`Line 170-173` and `Figure 2`), with 100B used in Round 1 and 120B mixed with other cross-modal data in Round 2. In the new experiments, we sampled **new 50B tokens** from LucaOne, bringing the total to 270B single modal data. These data **do not contain any repeating sequences**. Therefore, BSM and BSM-single share the same 220B single modal data, with the difference being that BSM used about 50B of cross-modal data, while BSM-single used about 50B of extra new single modal data. The data used in each round is summarized in the Table 2 below.

---

> ### Author Response · Authors · 2024-11-25
> **Response to Address Reviewer kPoi's Additional Questions (2/4)**
>
> Table 2:
> |                | Single modal data | RefSeq | Gene related sequence | Web |
> |----------------|---------|---------|---------|------------|
> | Round 1        | 100B    | /    | /       | /    |
> | Round 2         | 120B  | 24B    | 12B    | /         |
> | Round 3         | /       | 1.3B     | 1.3B    | 13.3B     |
> | BSM-single     | 270B      | /       | 13.3B   | /          |
> > 2. Why is the phase 3 data so effective? Phase 3 (the web data) seems to have a dramatic impact in a relatively few number of tokens.... I'm concerned that what you're showing isn't that "varied forms of data help", but rather that "pre-training on a set of sequences similar to those we find in the test set improves performances". While you've convinced me you're not leaking labels, I'm still concerned that your model is seeing very similar sequences in the pre-training data particularly in phase 3. The val ppl also decreasing rapidly may help point that phase 3 data actually does help the model learn something useful, but I'm not sure you can quite say this without a more rigorous exploration of what is in the validation data on how it affects different parts of it (e.g. does it decrease the ppl in only the protein portion of the val or uniformly across them)?. Perhaps I'm misunderstanding something about this and would appreciate some clarification on whether this is actually happening in practice or whether my concern is unfounded. I think this is worth discussion in much more detail in the main paper (what aspect of the web data is so effective and why are we sure our gains aren't from accidents leakage). Additionally, assuming that the web data is fine, why even bother with the first 2 phases then?
>
> Regarding these concerns, we provide the following explanation.
>
> 1. **Let's first talk about valid data**.
>   - Our valid data is single-modal data comes from LucaOne [1], which is randomly split from 800B tokens containing 169,861 species, while our web data only has 33 million tokens. We believe **there is no correlation or bias between the web data and the valid data**, so the results of the valid data in Figure 6 demonstrate that Round 3 can significantly improve the model's performance.
>
>   - To address your question, "does it decrease the PPL in only the protein portion of the validation data or uniformly across all of them?", we have provided additional data below. We divided the valid data into protein-only and gene-only subsets. By comparing the differences in the Round 2 and Round 3 checkpoints across these three types of valid data, it is clear that **both the protein and gene valid loss decreased through Round 3** training. In other words, the cross-modal data improved the representation for both modalities.
>
> Table 3:
> | Round | Overall Valid Data | Protein Valid Data | Gene Valid Data |
> |-------|--------------------|--------------------|-----------------|
> | Round2 | 1.2769             | 1.3019             | 1.2727          |
> | Round3 | 1.2287             | 1.2588             | 1.2236          |
>
> 2. **Analysis from downstream task test data**: As shown in the table in our response to Q1, both R2 and R3 data improve the model's performance. It's not just the web data—R2 shows a significant improvement, approaching the performance of other billion-scale models , and R3 continues to show further improvement. We believe the results on the valid data demonstrate that web data contributes to enhancing the model's performance in all single-modal data, which is consistent with the improvements observed in downstream tasks. When combining both results, they validate the benefits of web data.
>
> 3. **The Role of Cross-Modal Data in Round 2 and Round 3**:
>
> - In multi-modal models (such as text-image models), there are two commonly used types of data beyond single-modal data: image-text **pairs** and **interleaved data** (which consists of sequences of multiple images interspersed with text). The latter has gained increasing attention in recent research and has been widely shown to be effective [3, 4]. As an extension of image-text pairs, the interleaved format not only covers a broader range of data but also captures longer and more complex relationships.
>
> - We introduced these two types of data in BSM. In Round 2, we introduced **gene-protein pair data**, which **establishes semantic links between genes and proteins**, enhancing the model's learning efficiency across different modalities. In Round 3, we used **interleaved bio-sequence data obtained from the web**, which consists of sequences of proteins interspersed with genes. This data allows the model to **learn more complex and broader relationships between sequences over longer contexts, improving its performance on complex multi-modal tasks**.
>
> - Each type of data makes a unique contribution, and none is dispensable. By combining and leveraging these data types effectively, the model gains stronger and more comprehensive capabilities.

---

> > ### Author Response · Authors · 2024-11-25
> > **Response to Address Reviewer kPoi's Additional Questions (3/4)**
> >
> > 4. Why still need first 2 phases?
> >
> > - **Data Size Differences**: The amount of web data is very limited for bio sequences. We extracted all sequence combinations (sequences count ≥ 2) from FineWeb-Edu [5], but we only obtained 33 million tokens. While acquiring this data from larger web crawl datasets (e.g., CommonCrawl) could increase the data size, it would still be much smaller compared to datasets derived from sources like RefSeq or Gene-related sequences. (For today’s work, we used data from only a subset of species, with 8.3B tokens from RefSeq and 9.2B tokens from Gene-related sequences before upsampling.) Given the massive data size disparity, web data was not suitable for inclusion in Round 1&2.
> >
> > - In Round 1, a foundation for each modality was established. Experimental results have also shown that **the pair data in Round 2 positively impacts model performance**. And this type of data has not been fully exploited yet—e.g., gene-protein interaction pairs remain unexplored.
> >
> > - In Round 3, we trained the model on all high-quality cross-modal data together. The annealing process in the final training stage has been widely shown to improve model performance. As highlighted in Llama 3.1 (Section 3.1.3, 3.4.3) [6], upsampling and annealing on small amounts of high-quality data can significantly boost the performance of pre-trained models.
> >
> > Based on all the above information, we ultimately constructed a three-phase training process.
> >
> > > 3. I'm still unclear on the mix of phase 1 data used for training and eval. What's the mix of DNA:RNA:Protein, how was this data sampled? (uniformly at random? Sampling specific chromosomes/genes/ etc for some set of species?) I assume the authors also did some filtering/selection to make sure the val data didn't show up in the training data for each different phase, was the some process as the test data filtering used? ...To get your same performance gains is sampling uniformly at random from the same databases sufficient, or was there more that went into the sampling? Are they gains form each phase really from the multi-modal aspect of them or did they just happen to match the distribution of DNA:RNA:Protein better than your phase 1 data?
> >
> > 1. **Gene:Protein Ratio and sampling details**. LucaOne [1] released about 800B tokens of data containing 169,861 species, with a gene:protein ratio of about 6:1. We explicitly included 10B human DNA sequences (which is consistent with many studies that often use human DNA as training data [8][9]), and the rest of the sequences were randomly sampled. The final dataset totaled 220B tokens while maintaining the same gene:protein ratio. We will include this information in the paper.
> >
> > 2. **Reliability of valid data**. The valid data we used is from LucaOne, which was split from 800B tokens. We used the same valid data across all three rounds, with the only modification being the removal of sequences that appeared in the cross-modal training data. Given that the **valid data is sampled from an original dataset with both a much larger volume and a greater number of species than our cross-modal data**, we believe the **valid data is sufficiently diverse and that its overlap with the cross-modal data is minimal**, making it reliable for guiding our training.
> >
> > 3. **Reproductivity of BSM**. Although we do not have enough experimental data to quantify the effect of including human DNA, we believe its impact is minimal due to its relatively small proportion (10B vs. 220B tokens). We believe that pure random sampling would not significantly affect the results, but following our setting of including human data while randomly sampling other sequences can reliably reproduce our results. Additionally, our task does not involve any human-specific tasks.
> >
> > 4. **We want to emphasize that the gains come from using multi-modal data**.
> >
> > - In Round 1, we followed LucaOne with a gene:protein ratio of approximately 6:1, and the proportions of other cross-modal data are listed in below Table 4. **RefSeq and GeneRelatedSeq are the dominant cross-modal data, and their ratios are similar to those in Round 1**.
> >
> > - As demonstrated in above Table 3, **BSM does not improve only one type of data (e.g., proteins) but enhances performance for both genes and proteins**. While different ratios may influence performance on various downstream tasks, they are not the primary factors driving the observed gains.
> >
> > - The gains come from the introduction of new data types: cross-modal pair data and interleaved data. Pair data establishes relationships between two modalities, while interleaved data builds longer and more complex relationships, enhancing performance on complex multi-modal tasks.
> >
> > Today’s baseline methods, except for LucaOne, only incorporate DNA (e.g., Evo) or only protein (e.g., ESM). Determining the optimal ratio for training a better general bio model for both gene and protein is a worthwhile research question, which we leave for future work.

---

> > > ### Author Response · Authors · 2024-11-25
> > > **Response to Address Reviewer kPoi's Additional Questions (4/4)**
> > >
> > > Table 4:
> > > |  | Gene:protein|
> > > | -------------- | ----- |
> > > | Refseq        | 8:1   |
> > > | GeneRelatedSeq | 5:1  |
> > > | web           | 1:1   |
> > >
> > > ----
> > > **Reference**
> > >
> > > [1] LucaOne: Generalized Biological Foundation Model with Unified Nucleic Acid and Protein Language
> > >
> > > [2] Sequence modeling and design from molecular to genome scale with Evo, Science 2024
> > >
> > > [3] Gemini: A Family of Highly Capable  Multimodal Models
> > >
> > > [4] Chameleon: Mixed-Modal Early-Fusion Foundation Models
> > >
> > > [5] The FineWeb Datasets: Decanting the Web for the Finest Text Data at Scale
> > >
> > > [6] The Llama3 Herd of Models
> > >
> > > [8] DNABERT-2: Efficient Foundation Model and Benchmark For Multi-Species Genome
> > >
> > > [9] HyenaDNA: Long-Range Genomic Sequence Modeling at Single Nucleotide Resolution
> > >
> > > &nbsp;
> > >
> > > We sincerely appreciate your review. If you have any further questions or concerns, please feel free to contact us at any time.
> > >
> > > Best regards,
> > > All Authors

---

> > > > ### Comment · Reviewer_kPoi · 2024-11-27
> > > > **Response to follow up**
> > > >
> > > > Thank you for your response!
> > > >
> > > > The authors responses and addition are such that I'm comfortable raising my score from a 3 -> 5. I can see a path to a 6 (still 1 main concern for that, detailed below) but  even in an ideal case I think that's about as high I could conceivably go for this iteration of the work.
> > > >
> > > > ## Current Concern
> > > > With your current experiments, I believe I could be convinced to increase my score higher than a 5. The main concern I would need the authors to address remains the same:
> > > > 1. How do you I know you're not leaking data.
> > > >
> > > > I understand the authors claim that most of their performance gain comes from the multi-modal data, and I think they're right to an extent, but my concern remains since for example I can imagine the following failure mode: The test data contains protein sequence X. The web data contains protein sequence X' (possibly from a different specie) where the edit distance from X is small but non-zero. In my opinion this would constitute data leakage since many species have essentially the same protein with minor changes in regions of less consequence and I don't think this is accounted for in the current setup. I can image multiple such paths  (e.g. having the DNA sequence of X in the training set and the protein sequence of X in the test set, especially since the claim is that their model likely establishes semantic links between the modalities (which I think is likely true)). If the authors can convince me that whatever gains they get from the web data are not due to anything like this or that I am fundamental mistaken in my assumptions somewhere about this, I would find it justified to raise my score to at least a 6.
> > > >
> > > > ## Follow ups
> > > > In general, I feel like the most compelling part of this work is the authors data preparation (the multi-modal aspects and the phases for training). I think it seems clear that the authors pre-training has something about it that lets smaller models at least get    at least similar performance to significantly larger models (probably some mix of data quality and the different modalities like the authors suggest), and that seems genuinely useful and non-trivial in my opinion.
> > > >
> > > >  While I am genuinely enthusiastic about this idea and can see myself giving a paper on this topic a high score, the experiments I would need to see for that would probably be some mix of the following (in descending order of importance in my opinion).:
> > > > * Showing that your training pipeline scales to much larger models and that the performance gains over similar sized models but trained on uni-modal data stay the same. E.g Does this quality pre-training still scale to larger models, or do you hit limit where there's not enough clean data to improve performance on the large models? If the best this smarter pre-training data selection can do is let smaller models ~match the much larger models on some tasks, this severely limits how impactful this work would be and is a large concern I have with the current work.
> > > > * A thorough ablation on the web data to ascertain what exactly about it seems to be helping so much in such a relatively small amount of time. The web data seems very impactful relative to it's size, identifying what specifics aspect of it seems to help so much would therefore help us understand why in particular this work would be impactful and potentially help in the future design of biological pre-training data. (E.g. if the claim is that this data is higher quality/less noisy, what exactly does this mean? Less repetitive/lower entropy sequences? Sequences upsampled due to some known association with diseases? Etc).
> > > > * Include some DNA prediction tasks as well. your tasks are currently mainly on proteins with some RNA, and DNA seems to be a relatively  large portion of the pre-training data. You've convinced me your validation data is reasonably correlated to protein/RNA tasks, but this doesn't necessarily follow to DNA as well.
> > > >
> > > >
> > > > Naturally this represents a considerable amount of work and is not something that can be reasonably expected during a rebuttal period and as such I don't expect any work of this sort for this discussion window, however is the authors have some pre-existing work/additional data that would address these concerns or able to address them with the current work, I would absolutely be open to hearing about it.
> > > >
> > > > Regardless of the outcome, I encourage the authors to continue this line of work and look forward to reading the final product.

---

> > > > > ### Author Response · Authors · 2024-11-29
> > > > > **Response and Thanks to kPoi**
> > > > >
> > > > > We sincerely appreciate your recognition of our work and your consistent sense of responsibility. To address your concern regarding the potential data leakage from the web data:
> > > > >
> > > > > 1. We split the test data into two subsets based on an edit distance thresholds of 20%. Our experiments show that the **performance gain of R3 over R2 is unrelated to the similarity between test data and web data**, proving that the gains are not due to test sequences (or similar sequences) being present in the web data. '≤ 20%' means that the web data contains sequences that are similar to this test set."
> > > > >
> > > > > |                  | Protein Fitness(total/≤ 20%/>20%),count:1631/157/1474 | ProtLoc(total/≤ 20%/>20%),Count: 1131/109/1022 | ncRNA Fitness(total/≤ 20%/>20%),Count: 4566/103/4463 |
> > > > > |------------------|-----------------|---------------|------------|
> > > > > | BSM-R3 | 0.423(+5%)/0.419(+2.2%)/0.428(+9.7%)   | 0.9685(+3%)/0.9619(+3%)/0.9692(+3%) | 0.256(+7.1%)/0.250(+8.7%)/0.263(+6.9%) |
> > > > > |        BSM-R2         |        0.403/0.410/0.390         |    0.9401/0.9334/0.9408    |     0.239/0.230/0.246
> > > > >
> > > > > 2. We have already explained that the valid data is randomly split from 800B single-modal tokens, with minimal overlap with the web data(33M). It has been verified that **web data can significantly reduce valid loss, proving the significant impact of interleaved data**.
> > > > > 3. Today's LLMs perform data decontamination because their pretraining data contains rich information, which may include the solution and answers to math problems, or code for code tasks. However, while we use web data, we only extract biological sequences from it. As far as we know, all bio-sequence models today do not perform data decontamination. The top three tasks where R3 outperforms R2 are (ncRNA fitness, Protein fitness, ProtLoc), and the labels for these tasks are numerical scores that did not appear in the pretraining data. Our above experiments also show that there is no inherent relationship between performance gain and edit distance.
> > > > >
> > > > > For other follow-up questions, we are pleased to explain as much as we can.
> > > > > > Showing that your training pipeline scales to much larger models and that the performance gains over similar sized models but trained on uni-modal data stay the same.
> > > > >
> > > > > We apologize that we currently do not have enough resources to conduct such experiments. However, by comparing the validation curves of the 110M and 270M models in Figure 6, as well as the downstream task performance, we can demonstrate that scaling up still improves the model's performance.
> > > > >
> > > > > Scaling further with more types of multi-modal data and model sizes is something we believe holds significant potential and is worth pursuing. However, it remains a major challenge given our available resources. The value of our work lies in demonstrating the importance and potential of this direction. Currently, we have only explored a small portion of RefSeq and NCBI Gene-related Sequence data, but there are many other available cross-modal data sources. Therefore, there is still vast untapped potential both in terms of the types and volume of multi-modal data.
> > > > >
> > > > > Cross-modal data, especially interleaved data in longer contexts, has already been proven effective in many multi-modal scenarios. We hope to see more attention and work focused on multi-modal models in the bio domain.
> > > > >
> > > > > > A thorough ablation on the web data to ascertain what exactly about it seems to be helping so much in such a relatively small amount of time.
> > > > >
> > > > > We agree that the role of web data is significant relative to its volume. However, if you look at the added table for Q1, you'll see that the performance gain from Round 2 is actually larger than that from Round 3 (e.g., ncRPI Round 2 gain: 0.9216 → 0.9422, Round 3 gain: 0.9422 → 0.9494). The gain in Round 3 should be compared to BSM-R2, rather than BSM-single-R3.
> > > > >
> > > > > We believe web data is helpful due to **the role of interleaved data**, which is a key aspect of current research on multi-modal models. For example, single-modal data could be an image of a cat, and pairwise cross-modal data could be the phrase "This is a cat" paired with an image, connecting the two modalities. Interleaved data, on the other hand, might be a webpage discussing various pets, with text and images interleaved about different animals. **In longer contexts, such connections are more indirect, helping the model capture long-term relationships**. While interleaved data is much rarer, its impact is significantly greater. Our web data could be a webpage that discusses a broad biological topic and lists related bio sequences. By "high quality," we mean that this interleaved data is important but scarce, which is why we perform upsampling to increase its weight.

---

> > > > > > ### Author Response · Authors · 2024-11-29
> > > > > > **Response and Thanks to kPoi 2**
> > > > > >
> > > > > > > Include some DNA prediction tasks as well. your tasks are currently mainly on proteins with some RNA, and DNA seems to be a relatively large portion of the pre-training data. You've convinced me your validation data is reasonably correlated to protein/RNA tasks, but this doesn't necessarily follow to DNA as well.
> > > > > >
> > > > > > Thank you for your suggestion. We followed two works most relevant to ours: Evo and LucaOne. Evo does not include a DNA task, while LucaOne has a DNA task for predicting genus taxon (GenusTax). We didn't use it because LucaOne's pretraining includes Protein Taxonomy and Gene Taxonomy labels. We added the NucleotideTransformer[1] Benchmark for your reference. We will leave additional work for the future.
> > > > > >
> > > > > > | Tasks/Models | DNABERT-2 | BSM-110M | LucaOne |
> > > > > > |-------------|-----------|----------|---------|
> > > > > > | NT          | 0.678833  | 0.699278 | 0.664916 |
> > > > > >
> > > > > > [1] The Nucleotide  Transformer: Building and evaluating robust foundation models for human genomics. bioRxiv, 2023.
> > > > > >
> > > > > > We greatly appreciate your response and recognition; it means a lot to us.

---

> > > > > > > ### Comment · Reviewer_kPoi · 2024-12-03
> > > > > > > **Remaining Concerns on Data Leakage**
> > > > > > >
> > > > > > > I thank the Authors for their response.
> > > > > > >
> > > > > > > I don't think the current experiments are quite enough to convince me of the lack of data leakage, I need more details on how the sequences were split (how was the edit distance calculated? Bespoke alignment with smith-waterman or Needlrman-Wunch? Or was this using a protein blast/MMseq to identify sequence similarity)? Depending on some of the implementation details (e.g. substitution matrix used, gap penalties, etc) it might still be possible under the authors setup to have sequences that have a far edit distance but be very similar semantically.  Furthermore, if I understand correctly the split with similar sequences ('<20%') is relatively small in all tasks, making it difficult to actually say anything with confidence. Perhaps an easier way to show lack of contamination is using a method like MMSeq to search for homologs across the training data and the test set for each tasks for all the data modalities.
> > > > > > >
> > > > > > > Additionally, I'll point the authors to the following details from protiennet that I think succinctly explain some of the difficulties with  data contamination in protein data (even when the labels aren't being leaked, evolutionarily related sequence might be): https://github.com/aqlaboratory/proteinnet/blob/master/docs/splitting_methodology.md. This type of contamination is my specific concern, but the authors have the added difficulty of considering multiple modalities and possibly having leakage across modalities that makes this even more complicated as compared to just trying to adequately split protein sequences. This applies to every phase of the data, so I would encourage the authors to do some similarity analysis along  for each phase (possibly even between modalities in each phase), and if necessary in future iterations consider splitting their data into train/val based on sequence similarity.
> > > > > > >
> > > > > > > As for the rest of the authors response, I believe it to be on the right track, but is still insufficient. For example, I believe that the presence of interleaved data being a key component to the web data is a reasonable hypothesis, but there are no ablations or further analysis on the types of sequences in the phase 3 data that I would need to see to be confident about such a claim, and having something of that sort would very much improve the quality of the paper.
> > > > > > >
> > > > > > >  I thank the authors for engaging in discussion and look forward to reading the final work.

---

> ### Author Response · Authors · 2024-12-03
>
> I would like to emphasize that our two evaluation tasks, the **zero-shot protein fitness prediction task and the zero-shot ncRNA fitness prediction task, are both zero-shot evaluations**. In these tasks, we directly compute the Spearman correlation between the test sequence likelihood from the model and the fitness score. **For zero-shot tasks like these, the model has never been fine-tuned on downstream tasks, and the model has never seen the label at any data at any stage**, whether during pretraining or fine-tuning. Therefore, **there is no possibility of any leakage**.
> Our experimental results demonstrate significant improvement in both R2 and R3 for these tasks, proving the effectiveness of cross-modal data.
>
> We use edit distance because, in your description, you used it to measure sequence similarity. For the calculation of edit distance, we directly compute the minimum number of edit operations required to transform one sequence into another, without applying any specific sequence alignment technique. Based on this approach, the "<20%" threshold is not considered "relatively small" but is actually an appropriate threshold for our analysis.
>
> To clarify this, we can provide a table that summarizes the sequence count at various edit distance thresholds. This threshold ensures that the test set is appropriately divided. **If the threshold is too small (e.g., 10% or 15%), the number of similar sequences would be too few (<100), which may lead to unreliable statistics**. On the other hand, **we avoid using too large a threshold (e.g., 30%) because we believe smaller thresholds more strictly define a subset that closely resembles the web data. If there were indeed a leakage issue as you suggested, the performance gain would be more apparent for this subset relative to the others; however, the results show that this is not the case.**
>
> | Task     | Edit Distance ≤ 5% | Edit Distance ≤ 10% | Edit Distance ≤ 15% | Edit Distance ≤ 20%  | Edit Distance ≤ 30%  | Edit Distance ≤ 50%  |
> |----------|-----------|--------------|-------------------|----------|-----------|--------------|
> | ProtLoc(1131) | 1| 11 | 29 | 109 | 497 | 613 |
> | Zero-shot Protein Fitness (1631) | 2 | 6 | 14 | 157 | 558 | 1051 |
> | Zero-shot ncRNA Fitness (4566) | 0 | 9 | 19 | 103| 924 | 3107 |
>
> Thus, by using a statistically meaningful subset (with count >100) and a stricter similarity measure (<20%), we ensure that although the subset may contain sequences very similar to those in the web data, the performance gain observed does not exceed that of other data subsets.
>
> While you mentioned the use of other similarity measurement tools, **we believe that using a strict edit distance threshold of <20% still defines a reasonable and meaningful subset with strong correlation to web data. The results derived from this strict criterion are reliable and support that our performance gains are not due to data leakage**.
>
> Based on all the evidence provided above, we firmly believe that our contributions, particularly the effective use of cross-modal data, are solid and the research significance is substantial.

---

> > ### Author Response · Authors · 2024-12-03
> >
> > I may have a better understanding of why the reviewer raised concerns about data leakage, and I believe there may be some misunderstanding here. Please let me know if there are any inaccuracies in my interpretation below.
> >
> > Regarding the GitHub link you provided, ProteinNet is a dataset designed for assessing protein structure prediction, with the input being protein sequences and the label being the corresponding protein structures. It is a **task-specific dataset**, split into train/valid/test sets. In this type of **task-specific fine-tuning, data leakage could occur if label information from the training data leaks to the validation/test data via similar sequences**. However, this is **fundamentally different from our current pretraining and evaluation process across multiple downstream tasks**.
> >
> > In our pretraining, we **only learn next-token prediction based on sequences**. **No labels from downstream tasks are involved**—whether they are protein structures, classification labels, or anything else. This is why we mentioned that current **pretraining methods** on biological sequences do not involve data decontamination.
> >
> > **We have two types of downstream tasks: zero-shot and fine-tuning**. For zero-shot tasks, there is absolutely no possibility of leakage, as we explained above, since the model has never seen any labels at any data at any stage. Fine-tuning tasks, on the other hand, are widely used tasks that are completely unrelated to our pretraining process. Our method simply incorporates cross-modal data during the pretraining phase, with no risk of data leakage introduced by our approach.

---

### Author Response · Authors · 2024-11-22

Dear ACs and Reviewers,

Thank you very much for your thoughtful and constructive feedback. We would like to highlight the core contributions and significance of our work, which may have been overlooked.

Existing work on biological sequence models has primarily focused on scaling model size to the billion-parameter level. However, insufficient attention has been given to exploring diverse data, which is crucial for models to acquire diverse capabilities. Different types of bio seq data, i.e., DNA, RNA, and proteins, are intrinsically interconnected within a genetic flow. Our work proves that learning from multiple modalities and mixed-modal data significantly enhances learning efficiency and improves both single and mixed-modal representations.

Through experiments on BSM-110M/270M models, we demonstrate that incorporating cross-modal data allows the model to achieve performance comparable to billion-scale models. We also validate the scalability of our approach, showing that further increasing model size continues to enhance performance. This provides strong support for future efforts in scaling cross-modal data and model size to develop a stronger bio seq model.

We addressed the main issues raised by reviewers and updated the PDF:
- **Added an ablation study on cross-modal data**: Compared models trained on single-modal vs. with cross-modal data under the same token budget, showing that performance gains come from cross-modal data, not longer training.(Added in `Section 3.5`)
- **Added a baseline of ESM-150M**: Demonstrating that BSM significantly outperforms baseline models of the same size as ours. (Added in `Section 3.6`)
- **Added details about the data mix.**

&nbsp;

Best regards,
All Authors

---

### Meta-Review · Area_Chair_cxDy · 2024-12-20

**Metareview:**

This paper introduces BSM, a small mixed-modal biological sequence foundation model that seeks to enhance cross-modal representation and task performance. The authors argue that leveraging cross-modal data enables smaller models to achieve performance comparable to or exceeding larger models, emphasizing data quality over sheer model size. BSM is evaluated on various single-modal and mixed-modal tasks, where it outperforms several billion-parameter models despite having only 110M–270M parameters.

**Additional Comments On Reviewer Discussion:**

Reviewer concerns include the lack of sufficient ablation studies to isolate the contributions of cross-modal data, inadequate discussion on data splitting and potential leakage, and limited exploration of how the model benefits specific task categories. Presentation issues such as repetitive sections, low-quality figures, and unclear experimental details were also brought up. While the integration of cross-modal data is a promising direction, reviewers found that the current submission does not establish the novelty and reliability of its contributions, and these concerns persisted after the rebuttal.

---

### Decision · Program_Chairs · 2025-01-22

Reject